# Repeated Augmented Rehearsal: A Simple but Strong Baseline for Online Continual Learning

**Yaqian Zhang**
University of Waikato
New Zealand
yaqianz@waikato.ac.nz

**Bernhard Pfahringer**
University of Waikato
New Zealand
bernhard@waikato.ac.nz

**Eibe Frank**
University of Waikato
New Zealand
eibe@waikato.ac.nz

**Albert Bifet**
University of Waikato
New Zealand
LTCI, Télécom Paris, France
abifet@waikato.ac.nz

**Nick Jin Sean Lim**
University of Waikato
New Zealand
nick.lim@waikato.ac.nz

**Yunzhe Jia**
University of Waikato
New Zealand
alvin.jia@waikato.ac.nz

## Abstract

Online continual learning (OCL) aims to train neural networks incrementally from a non-stationary data stream with a single pass through data. Rehearsal-based methods attempt to approximate the observed input distributions over time with a small memory and revisit them later to avoid forgetting. Despite their strong empirical performance, rehearsal methods still suffer from a poor approximation of past data's loss landscape with memory samples. This paper revisits the rehearsal dynamics in online settings. We provide theoretical insights on the inherent memory overfitting risk from the viewpoint of biased and dynamic empirical risk minimization, and examine the merits and limits of repeated rehearsal. Inspired by our analysis, a simple and intuitive baseline, repeated augmented rehearsal (RAR), is designed to address the underfitting-overfitting dilemma of online rehearsal. Surprisingly, across four rather different OCL benchmarks, this simple baseline outperforms vanilla rehearsal by 9%-17% and also significantly improves the state-of-the-art rehearsal-based methods MIR, ASER, and SCR. We also demonstrate that RAR successfully achieves an accurate approximation of the loss landscape of past data and high-loss ridge aversion in its learning trajectory. Extensive ablation studies are conducted to study the interplay between repeated and augmented rehearsal, and reinforcement learning (RL) is applied to dynamically adjust the hyperparameters of RAR to balance the stability-plasticity trade-off online. Code is available at https://github.com/YaqianZhang/RepeatedAugmentedRehearsal.

## 1 Introduction

Despite its recent success, deep learning largely relies on the assumption of independent and identically distributed (i.i.d.) data that can be repeatedly revisited during training. Non-i.i.d settings are challenging for neural networks due to catastrophic forgetting: previously learned knowledge can easily be overwritten when training on new data because this data may follow a different distribution [Li and Hoiem, 2017, Rebuffi et al., 2017, Delange et al., 2021]. Online continual learning (OCL or Online CL) studies how to enable deep learning in an online manner from a non-stationary data stream. As the data stream can be vast or even infinite, it is infeasible to store and shuffle the dataset for multiple epochs of training. Therefore, a fundamental assumption is that the data stream can only

be accessed one batch at a time and training is performed with a *single pass* over the data [Aljundi et al., 2019].

Experience replay (ER), also known as rehearsal [Chaudhry et al., 2019, Delange et al., 2021], is a key idea in OCL. It stores a subset of previously seen data $\mathcal{D}$ in a fixed-size memory $\mathcal{M}$ and revisits the memorized samples during training to mitigate forgetting of previous tasks. To update the model, a batch sampled from the memory is combined with the incoming batch from the stream to compute the gradient [Chaudhry et al., 2019]. Different variants of ER have been developed to improve memory management policies and representation learning, achieving state-of-the-art performance in a number of standard OCL benchmarks [Aljundi et al., 2019, Mai et al., 2021, Shim et al., 2021].

However, whether rehearsal is appropriate for continual learning, considering the risk of overfitting the memory when using data from the memory to directly contribute to the gradient computation, has been debated vigorously [Lopez-Paz and Ranzato, 2017, Chaudhry et al., 2019, Verwimp et al., 2021]. The potential for overfitting has motivated the development of constraint-based replay methods [Lopez-Paz and Ranzato, 2017], e.g., GEM and A-GEM, which use memory samples solely to *constrain* the gradient direction. However, there is empirical evidence suggesting that rehearsal-based methods consistently outperform methods that do not train directly on the memory [Chaudhry et al., 2019]. This indicates rehearsal on the memory does not necessarily prevent effective generalization, possibly due to the implicit regularization effect of incoming data [Chaudhry et al., 2019]. Nevertheless, recent work analyzing the loss landscape when applying rehearsal to offline continual learning finds that memory samples indeed provide a poor approximation of the loss landscape of past tasks, especially near a high-loss ridge. As a result, "instead of ending up *near* the high-loss ridge in perspective of the rehearsal memory, the solution in reality resides *on* the high-loss ridge for the training data" [Verwimp et al., 2021]. This latest finding poses the question of how to better approximate the loss surface of past data $\mathcal{L}(\mathcal{D}; \theta)$ with memory samples' loss $\mathcal{L}(\mathcal{M}; \theta)$.

To better approximate past data's loss surface, previous work studies which samples should be memorized for rehearsal. Instead, we examine the optimization process during rehearsal and study how to effectively perform rehearsal with the memorized samples. Focused on *online* CL, our study extends the previous understanding of rehearsal along two directions. First, we provide theoretical considerations that reveal two insights regarding the extent of overfitting to memory: it is a) related to the inherent attributes of the OCL problem concerned and b) varies across the different stages of continual learning. Second, we highlight the limits of applying rehearsal with multiple iterations—a trick used to maximally utilize the incoming batch [Aljundi et al., 2019]—and identify an underfitting-overfitting dilemma for online rehearsal.

Based on our analysis, we design a simple baseline to deal with the underfitting-overfitting dilemma in online CL problems, dubbed repeated augmented rehearsal (RAR), that can be easily integrated into existing rehearsal-based methods. Surprisingly, this simple baseline leads to a large performance boost for ER, as well as state-of-the-art ER-based approaches, across four OCL benchmarks. More importantly, the loss landscape analysis shows that RAR can help memory samples reliably approximate the distribution of past data and successfully avoids the high-loss ridge of past tasks. To better understand the behavior of RAR, we further investigate the interplay between repeated rehearsal and augmented rehearsal via an ablation study. We also propose a reinforcement learning-based method to dynamically adjust the hyperparameters of RAR and balance the stability-plasticity trade-off in an online manner.

## 2 Related Work

**Online continual learning**: We consider the online continual learning setting with a non-stationary (potentially infinite) stream of data $\mathcal{D}_t$: at each time step $t$, the continual learning agent receives an incoming batch of data samples $\mathcal{B}_t = \{\mathbf{x}_i, y_i\}_{i=1,...,|\mathcal{B}_t|}$ that are drawn from the current data distribution $\mathbb{P}(\mathcal{D}_t)$. The period of time where the data distribution stays the same is often called a *task* or *experience* in the continual learning literature. An abrupt change in the data distribution occurs when the task changes. The standard objective during training is to minimize the empirical risk on all the data seen so far:

$$\min_{\theta} \mathcal{R}(\theta) = \min_{\theta} \frac{1}{\sum_t |\mathcal{B}_t|} \sum_t \sum_{\mathbf{x}, y \in \mathcal{B}_t} \mathcal{L}\left(f_\theta(\mathbf{x}), y\right), \tag{1}$$

with loss function $\mathcal{L}$, the CL network function $f$, and its associated parameters $\theta$.

*Metrics*: A common metric is the end accuracy after training on $T$ tasks, defined as $A_T = \frac{1}{T} \sum_{j=1}^{j=T} a_{T,j}$, where $a_{i,j}$ denotes the model's accuracy on the held-out test set of task $j$ after training on task $i$. Other metrics are "forgetting" [Chaudhry et al., 2018], which is defined as $F_T = -\frac{1}{T-1} \sum_{i=1}^{T-1} \left( a_{T,i} - \max_{l \in 1...T-1} a_{l,i} \right)$ and the related metric "backward transfer" [Lopez-Paz and Ranzato, 2017]: $B_T = \frac{1}{T-1} \sum_{i=1}^{T-1} a_{T,i} - a_{i,i}$.

*Online setting*: A key difference between online and offline CL is that the latter assumes full access to the whole training data for the task that is currently being processed. Therefore, it allows training on each single task with multiple epochs (e.g., 70-200 epochs) [Rebuffi et al., 2017, Wu et al., 2019, Cha et al., 2021]. The online CL setting is more challenging because the agent can only access the current batch of incoming data and performs training with a single pass through the data.

**Experience Replay (Rehearsal)**: Chaudhry et al. (2019) propose experience replay (ER), which performs joint training on memory samples and incoming samples. A simple but strong baseline approach to sampling in ER is reservoir sampling Vitter [1985]. Aljundi et al. (2019) propose Maximally Interfered Retrieval (MIR), which retrieves the samples that will be most negatively impacted by the foreseen parameter updates. Shim et al. (2021) propose ASER, which selects samples to best preserve existing memory-based class boundaries. In terms of model training, Mai et al. (2021) propose to replace the cross-entropy loss with the supervised contrastive loss to learn a better representation. We consider all these variants of ER in our experiments.

*Augmentation*: In the standard i.i.d. setting, data augmentation is a widely used method to improve deep learning [Cubuk et al., 2020]. In the offline CL setting, Mai et al. [2021] use augmentation to construct a supervised contrastive loss and Bang et al. [2021] employ it together with an uncertainty-based memory management strategy. However, these papers apply augmentation together with other advanced techniques, and there is no ablation study on the effect of augmentation per se. Thus, it is unclear whether augmentation itself is beneficial to rehearsal or not, especially in the online setting.

*Repeated Rehearsal (Multiple Iterations)*: Vanilla online continual learning employs a single gradient update given an incoming batch of data. To maximally utilize the current incoming batch, Aljundi et al. [2019] propose to perform multiple gradient updates instead. Their experiment with CIFAR10 shows using MIR with five iterations leads to a 1.7% improvement in accuracy. In this paper, we systematically analyze the effect of multiple iterations on online rehearsal and provide theoretical and empirical insights on when this trick may improve or harm performance.

**Hyperparameter Tuning for OCL**: Hyperparameter tuning is a particular challenge in OCL due to the lack of a dedicated validation set and the constraint of a single pass through the data. Chaudhry et al. [2018] and Mai et al. [2022] employ a hyperparameter tuning protocol that uses an external validation data stream with a small number of tasks. Offline hyperparameter tuning is applied to this validation data with multiple passes to identify optimal values, which are then used for the actual online continual learning tasks. A limitation of this method is that it relies on external validation data.

## 3 Revisiting Online Rehearsal: Is Repeated Rehearsal a Good Idea?

We revisit rehearsal from two directions. First, while previous work demonstrates its strong empirical performance [Chaudhry et al., 2019] and provides conceptual analysis for *offline* CL [Verwimp et al., 2021], we focus on the *online* setting and provide theoretical insights through the lens of empirical risk minimization (ERM). Second, we examine the dynamics of employing rehearsal with multiple iterations. This has been proposed as a trick for online CL to maximally utilize the incoming batch already [Mai et al., 2022, Aljundi et al., 2019]; we investigate whether it is always better than rehearsal with a single iteration.

### 3.1 Empirical Risk Minimization in Online Rehearsal: a Biased and Dynamic Objective

For rehearsal in OCL, at each iteration $t$, a batch of data $\mathcal{B}_t$ is obtained from the incoming task, where $\mathcal{B}_t \sim \mathcal{D}_\mathcal{T}$ and $\mathcal{B}_t = \{\mathbf{x}_i, y_i\}_{i=1,...|\mathcal{B}_t|}$, and a batch $\mathcal{B}_t^\mathcal{M}$ is sampled from memory, where $\mathcal{B}_t^\mathcal{M} \sim \mathcal{D}_\mathcal{M}^t$ and $\mathcal{B}_t^\mathcal{M} = \{\mathbf{x}_i, y_i\}_{i=1,...|\mathcal{B}_t^\mathcal{M}|}$. The gradient-based update rule of ER is:

$$\theta_{t+1} = \theta_t - \frac{\eta}{|\mathcal{B}_t|} \sum_{\mathbf{x},y \in \mathcal{B}_t} \nabla \mathcal{L}\left(f_\theta(\mathbf{x}), y\right) - \frac{\eta}{|\mathcal{B}_t^\mathcal{M}|} \sum_{\mathbf{x},y \in \mathcal{B}_t^\mathcal{M}} \nabla \mathcal{L}\left(f_\theta(\mathbf{x}), y\right). \tag{2}$$

Given this update rule, we would like to establish the corresponding objective function, but this is not straightforward to derive because the memory is immediately updated after each incoming batch, which means the memory data samples $\mathcal{D}_{\mathcal{M}}^t$ changes all the time. For a widely used memory management policy, where the memory is updated using reservoir sampling [Vitter, 1985, Chaudhry et al., 2019], we prove in the appendix that the empirical risk for online ER follows Proposition 1.

**Proposition 1 (ERM for online rehearsal)**: Assume an incoming task stream $\mathcal{D}_{\mathcal{T}}$ and an initial memory set $\mathcal{D}_{\mathcal{M}}^0$ with different data distribution $\mathbb{P}(\mathcal{D}_{\mathcal{T}}) \neq \mathbb{P}(\mathcal{D}_{\mathcal{M}}^0)$. Assume further that the memory is updated at the end of each iteration using reservoir sampling. Then, Eq 2 implements unbiased stochastic gradient descent for the following loss function:

$$\min_{\theta} \mathcal{R}_t(\theta) = \min_{\theta} \sum_{\mathbf{x},y \in \mathcal{D}_{\mathcal{T}}} \mathcal{L}(f_\theta(\mathbf{x}), y) + \beta_t \lambda \sum_{\mathbf{x},y \in \mathcal{D}_{\mathcal{M}}^0} \mathcal{L}(f_\theta(\mathbf{x}), y), \tag{3}$$

where $\lambda := \frac{|\mathcal{D}_{\mathcal{T}}|}{|\mathcal{D}_{\mathcal{M}}^0|}$ and $|\mathcal{D}_{\mathcal{M}}^0|$ and $|\mathcal{D}_{\mathcal{T}}|$ are the memory size and incoming task data size respectively; $\beta_t := 1/(1 + \frac{2N_{cur}^t}{N_{past}^{\mathcal{T}}})$, and $N_{cur}^t = \sum_{i=1}^{i=t} |\mathcal{B}_i|$ denotes the number of samples of the current task that have been seen so far and $N_{past}^{\mathcal{T}} = \sum_{j=1}^{j=\mathcal{T}} |\mathcal{D}_j|$ denotes the number of samples pertaining to the tasks so far, excluding the current task. We immediately have $\beta_t \in (0, 1]$.

The proposition reveals several interesting properties:

- **Bias**: compared with the true objective in continual learning in Eq 1, the loss of online ER in Eq 3 is actually a biased approximation of the former, as it puts a different weight ($\beta_t \lambda$) on memory samples while the true objective treats all samples with equal weight. This bias, introduced by ER's objective function, can contribute to the risk of memory overfitting.

- **Problem-dependence**: given that $\beta_t \in (0, 1]$, the biased weight on the memory samples is mostly influenced by $\lambda$. In other words, the memory overfitting risk is related to an inherent property of the CL problem concerned: the ratio $\lambda$ between the current task data size and the memory data size.[1] While previous works extensively report empirical results on the influence of memory size, to our knowledge, we are the first to point out that the relative data size of an incoming task also plays an important role. Empirical evidence is provided in Section 6.3 to support this claim. With a 2k memory, performing rehearsal on the CORE50 dataset with a larger task data size ($\lambda = 6$) faces a high level of memory overfitting while the CLRS dataset with a smaller task data size ($\lambda = 1.12$) enjoys a lower risk of memory overfitting.

- **Dynamic**: ER in online CL optimizes towards a *dynamic* objective, which varies with each incoming batch $t$, as the weight on the memory sample depends on $N_{cur}^t$, and $N_{past}^{\mathcal{T}}$. This analysis shows that memory overfitting may be relatively slight for the first few tasks when $N_{past}^{\mathcal{T}}$ is small. As more tasks arrive, memory overfitting worsens as $\beta_t$ increases with $N_{past}^{\mathcal{T}}$. In the case of an infinite data stream, $\lim_{N_{past}^{\mathcal{T}} \to \infty} \beta_t = 1$, the memory weight is solely determined by $\lambda$.

### 3.2 Repeated Rehearsal: The Decaying Regularization Effect of Incoming Data

We now investigate whether performing multiple iterations is beneficial to rehearsal or not. We refer to applying multiple iterations in ER as "repeated experience replay" (Repeated ER) or "repeated rehearsal" and formalize it as follows: for each incoming data batch $\mathcal{B}_t \sim \mathcal{D}_{\mathcal{T}}$ from the data stream, we perform multiple gradient updates ($K$ in total) using stochastic gradient descent (SGD) or variants thereof. At each gradient update $k = 1, ...K$, a data batch $\mathcal{B}_{t,k}^{\mathcal{M}}$ is chosen from memory $\mathcal{M}$ and concatenated with the current incoming batch $B_t$ to perform a replay iteration as follows:

$$\theta_{t,k+1} = \theta_{t,k} - \frac{\eta}{|\mathcal{B}_t|} \sum_{\mathbf{x},y \in \mathcal{B}_t} \nabla \mathcal{L}\left(f_{\theta_{t,k}}(\mathbf{x}), y\right) - \frac{\eta}{\left|\mathcal{B}_{t,k}^{\mathcal{M}}\right|} \sum_{\mathbf{x},y \in \mathcal{B}_{t,k}^{\mathcal{M}}} \nabla \mathcal{L}\left(f_{\theta_{t,k}}(\mathbf{x}), y\right). \tag{4}$$

---

[1]The experiments in this paper mainly consider a balanced CL setting where incoming tasks have the same data size. For imbalanced CL cases, $\lambda$ is also dependent on different tasks and should be expressed as $\lambda^{\mathcal{T}}$. The finding remains the same.

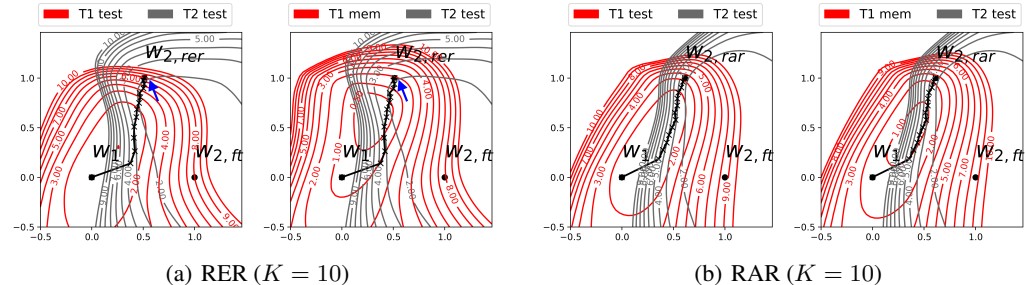

|     |     |
| --- | --- |
| (a) RER ($K = 10$) | (b) RAR ($K = 10$) |

Figure 1: Loss contours of RER and RAR. Memory data overfitting can be observed in (a) for RER but not in (b) for RAR. Note how the shape and position of the loss contour of "T1 test" differs from the "T1 memory" loss contour in (a). At the CL solution point, the test loss is 7.9 (left blue arrow) while the memory loss is only 2.1 (right blue arrow).

Note that when $k = 1$, this update rule provides an unbiased stochastic gradient for Eq 3; in the case of $k > 1$, it provides a biased gradient estimate as the same incoming batch is used for consecutive gradient updates. To study the influence of this biased gradient update in repeated rehearsal, we examine the internal dynamics of repeated rehearsal by studying the loss landscape.

**Memory Overfitting**: We compare the loss surfaces regarding the memory samples and the test data of past tasks during repeated rehearsal. Following the visualization method used in Verwimp et al. [2021], Mirzadeh et al. [2020], we examine the learning process on the first two tasks in the Split Mini-ImageNet dataset and plot the loss landscape in the 2D plane defined by three model parameter vectors[2]: the model $w_1$ obtained by training on the first task until convergence, the model $w_2$ obtained on the second task using experience replay, and the model $w_{2,ft}$ obtained after training on the second task using finetuning without replay. Verwimp et al. (2021) use this method to demonstrate the memory overfitting in ER for *offline* continual learning setting with 10 epochs. Our results show that applying ER in the *online* setting yields severe memory overfitting with 10 iterations (see Fig 1 (a)). In other words, increasing the number of iterations means the loss landscape of the rehearsal memory provides a poorer approximation of the loss landscape of previous tasks, as shown by the differences in the positions and the shapes between the left and right red contours in Fig 1 (a). As a result, the learning trajectory of RER avoids the high-loss ridge region for the memory data but goes right into the high-loss ridge region for the past tasks' test data.

**Regularization Effect of Incoming Data**: To investigate why repeated rehearsal may suffer from even more memory overfitting than vanilla rehearsal, we analyze the regularization effects of incoming data during repeated ER. To this end, we examine the training process given an incoming batch and compare the training loss on the memory batch and incoming batch with respect to memory iteration $k$ (see Fig 2 ). An interesting observation is that during the training session of a given incoming data batch, the decrease in the training loss on the incoming batch is much faster than the decrease in loss on the memory batches. One intuitive explanation is that the former is computed over a fixed batch during multiple iterations and the latter is computed over different memory batch samples. As a result, even though the incoming loss is larger than the memory loss at the start of a training session ($k = 1$), at later iterations (i.e., $k > 5$) the training loss of the incoming batch becomes $10 - 10^2$ times lower than that of the memory batch. This means that at this stage the regularization effect of the incoming data batch is greatly undermined: the joint training on the memory batch and the incoming batch becomes similar to training on memory only.

In summary, our findings imply that the performance of online rehearsal is constrained by the dilemma between overfitting locally and underfitting globally. Specifically, online rehearsal faces the challenge of underfitting of the large data stream but overfitting of a small memorized data subset. Applying repeated rehearsal ameliorates the former problem but aggravates the latter problem. Therefore, the performance gain from repeated rehearsal is quite limited.

---

[2]When training $w_2$ and $w_{2,ft}$, the model is initialized from $w_1$. The memory contains 100 samples/task (see Fig 1 (b) left and right). The 2-d coordinate system is built by orthogonalizing $w_2 - w_1$ and $w_{2,ft} - w_1$.

## 4    Repeated Augmented Rehearsal

To deal with the overfitting-underfitting dilemma, we explore a simple strategy, "repeated augmented rehearsal" (RAR), which combines repeated rehearsal with data augmentation. Consider a group of transforms $G$ that acts on the input space $\mathcal{X}$ and is invariant under function $f$, i.e., $f(g\mathbf{x}) = f(\mathbf{x}), g \in G, \mathbf{x} \in \mathcal{X}$. Given an incoming batch from the data stream, $\mathcal{B}_t$, multiple replay iterations are conducted using this batch. At each replay iteration $k$, a random memory batch $\mathcal{B}_{t,k}^{\mathcal{M}}$ is sampled and concatenated with the incoming batch. Then, a random transform $g_{t,k} \in G$ is sampled and applied to each data point $\mathbf{x}_i$ in the concatenated minibatch. The model parameters are updated as :

$$g_{rar} = \frac{1}{|\mathcal{B}_t|} \sum_{\mathbf{x},y \in \mathcal{B}_t} \nabla \mathcal{L}\left(f_\theta(g_{t,k}\mathbf{x}), y\right) + \frac{1}{\left|\mathcal{B}_{t,k}^{\mathcal{M}}\right|} \sum_{\mathbf{x},y \in \mathcal{B}_{t,k}^{\mathcal{M}}} \nabla \mathcal{L}\left(f_\theta(g_{t,k}\mathbf{x}), y\right).$$

Intuitively, the augmentation can help alleviate memory overfitting in two ways. First, we observe that applying augmentation on the incoming batch helps strengthen the regularization effect. As shown in Fig 2(b), the decaying regularization effect of incoming data is alleviated in RAR, as the loss of the incoming batch stays comparable to the loss of the memory batch during multiple iterations. Second, rehearsal on augmented memory batches can help to more accurately reflect past tasks' data distributions. With RAR, the loss landscapes of memory data and past tasks' test data become very similar (see Fig 1 (b)), which suggests that the model ends up in a part of the parameter space where the rehearsal memory approximates the past tasks' distribution well. Moreover, the continual learning solution identified with RAR avoids the high-loss ridge not only in the memory data loss landscape but also in the test data loss landscape.

Theoretically, we prove that augmented rehearsal reduces the generalization error in OCL. Specifically, assume the augmentation group $G$ is a compact topological group and follows a probability distribution $\mathbb{Q}$. Similar to **Proposition 1**, it can be easily proven that the augmented rehearsal gradient corresponds to unbiased SGD on an augmented empirical risk[3] (see **Proposition 3** in Appendix A):

$$\bar{\mathcal{R}}_t(\theta) = \sum_{\mathbf{x},y \in \mathcal{D}_{\mathcal{T}}} \int_G \mathcal{L}(f_\theta(g\mathbf{x}), y) d\mathbb{Q}(g) + \beta_t \lambda \sum_{\mathbf{x},y \in \mathcal{D}_{\mathcal{M}}} \int_G \mathcal{L}(f_\theta(g\mathbf{x}), y) d\mathbb{Q}(g). \tag{5}$$

This result shows that applying augmented rehearsal is equivalent to performing an averaging operation of the loss of rehearsal in Eq 3 over the orbits of a certain group that keeps the data distribution approximately invariant. In the standard i.i.d. learning setting, Chen et al. [2020] found that such an orbit-averaging operation can reduce both the variance and generalization error. Based on Eq 3 and Eq 5, we show that this theoretical benefit of using augmentation to boost model invariance is also applicable to rehearsal in continual learning. In fact, as discussed in Section 3.1, rehearsal in online CL has a biased empirical risk Eq 3, which leads to inherent memory overfitting and poor generalization ability. Thus, this benefit of augmentation in reducing generalization error is particularly important when applying rehearsal in online CL.

The modifications required for the RAR procedure are summarized in Lines 3 and 5 of Algorithm 1. It uses a general framework for ER-based continual learning that consists of three key components: sampling from memory, joint training on memory data and incoming data, and updating of the memory. As mentioned in the related work section, different ER variants have been proposed to improve these components. RAR can flexibly be combined with any of these ER variants, and we investigate the effectiveness of RAR on these different ER variants in the experiment section.

## 5    Reinforcement Learning-based Adaptive Repeated Augmented Rehearsal

There are two key components in RAR: repeated rehearsal and augmented rehearsal. The interplay of the two is determined by the number of memory iterations and the strength of augmentation. A key question is how to choose these hyperparameters. In general, hyperparameter tuning (HPT) still remains an unsolved challenge for online CL due to the single-pass assumption [Chaudhry et al., 2018]. Finding suitable RAR hyperparameters needs to account for the severity of memory overfitting

---

[3]Note that the theoretical analysis of the loss functions in Eq 3 and Eq 5 is also applicable to offline continual learning, which may be of independent interest. More discussion of the influence of augmentation in offline continual learning can be found in Appendix D.6.

Algorithm 1: RL-based RAR

---

$\mathcal{M}$ is the memory with fixed size,
$\mathcal{B}_t$ is the incoming batch from the current task,
$\theta$ are the parameters of the CL network,
$w$ are the parameters of the RL agent,
$K$ is the number of memory iterations,
$P, Q$ are the augmentation hyperparameters

1: **procedure** RAR($\mathcal{M}_t, \mathcal{B}_t, \theta_t, w_t$)
2:     $K_t, P_t, Q_t = SampleAction(w_t)$
3:     **for** $k = 1, ..., K_t$ **do**
4:         $\mathcal{B}_{t,k}^{\mathcal{M}} \sim MemRetrieval(\mathcal{M}_t)$
5:         $\mathcal{B}_{aug} \leftarrow aug(\mathcal{B}_{t,k}^{\mathcal{M}} \cup \mathcal{B}_t, P_t, Q_t)$
6:         $r_t \leftarrow ComputeReward(\mathcal{B}_{aug}, \theta_{t,k})$
7:         $\theta_{t,k+1} \leftarrow SGD(\mathcal{B}_{aug}, \theta_{t,k})$
8:     **end for**
9:     $\mathcal{M}_{t+1} \leftarrow MemUpdate(\mathcal{M}_t, \mathcal{B}_t)$
10:     $w_{t+1} \leftarrow UpdateRL(r_t)$
11: **end procedure**

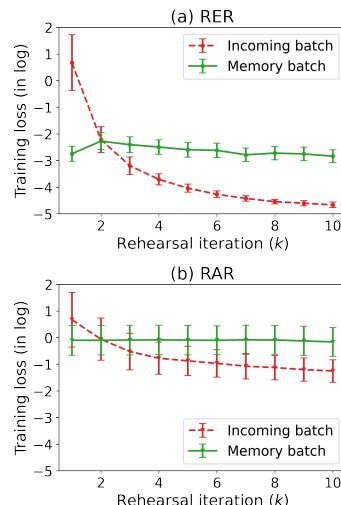

Figure 2: Memory loss vs. incoming loss.

and poses extra challenges. In particular, as shown in the ERM analysis, the extent of memory overfitting is related to the CL problem features (e.g., task data size and memory size) and also varies at different training stages of the continual learning process. To automatically select suitable RAR hyperparameters for the different CL problems and different training stages, we propose to use reinforcement learning to adaptively adjust the hyperparameters (see Algorithm 1).

In particular, we design the hyperparameters of RAR as the action space and use the training statistics as the reward (see lines 2, 6, and 10 in Algorithm 1). A major challenge of applying RL in online HPT is sample efficiency. The exploration horizon (i.e, training steps) in the OCL environment is quite limited due to the constraint of a single pass through the data and poor action choices (undesirable hyperparameters) may lead to a bad gradient update step and hurt the OCL training process. To address the sample efficiency issue, we employ the multi-armed bandit framework and apply bootstrapped policy gradient (BPG) [Zhang and Goh, 2019]. The key idea of BPG is to incorporate prior knowledge to bootstrap the policy gradient to achieve stable and fast convergence with limited samples. To obtain prior knowledge in OCL problems, we use the training accuracy on the memory batch as the overfitting feedback, as a higher training memory accuracy suggests a higher chance of memory overfitting. Reward is defined as the distance between the current memory accuracy and target memory accuracy. Compared against a target memory accuracy (e.g., 0.9), the current memory accuracy is used to indicate whether the current choice of rehearsal iteration or augmentation causes too much memory overfitting and then the action selection probability is adjusted following BPG (see Appendix C.2 for more RL design and implementation details). The algorithm details of applying BPG as a specific RL method for hyperparameter tuning is summarized in Algorithm 2 of Appendix C.2).

## 6 Experiments

### 6.1 Experiment Setup

**Baseline**: We apply RAR to four ER-based continual learning algorithms: ER [Chaudhry et al., 2019], MIR [Aljundi et al., 2019], ASER [Shim et al., 2021], and SCR [Mai et al., 2021]. We also compare it with other continual learning methods, including the regularization-based method LWF [Li and Hoiem, 2017] and the constrained optimization-based method A-GEM [Chaudhry et al., 2018].

**Dataset**: Four CL benchmarks are used in the experiments: Seq-CIFAR100 (20 tasks), Seq-MiniImageNet (10 tasks) Vinyals et al. [2016], CORE50-NC (9 tasks) Lomonaco and Maltoni [2017] and CLRS25-NC (5 tasks) Li et al. [2020](see Appendix B for more details). Additionally, we also investigate ER and RAR on the large-scale ImageNet-1k dataset in Appendix D.3.

Table 1: Accuracy on four OCL benchmarks with 2k and 5k memory. The performance boost of RAR over ER and ER variants is shown.

| | SEQ-CIFAR100 | | SEQ-MINI-IMAGENET | | CORE50-NC | | CLRS25-NC | |
|---|---|---|---|---|---|---|---|---|
| | 2K | 5K | 2K | 5K | 2K | 5K | 2K | 5K |
| FINETUNE | $3.2 \pm 0.1$ | | $4.3 \pm 0.8$ | | $7.7 \pm 0.2$ | | $6.5 \pm 0.9$ | |
| LWF | $8.7 \pm 0.5$ | | $10.9 \pm 0.5$ | | $9.6 \pm 0.3$ | | $12.4 \pm 2.2$ | |
| AGEM | $8.5 \pm 0.4$ | $9.2 \pm 0.2$ | $11.6 \pm 0.1$ | $13.1 \pm 0.3$ | $18.6 \pm 0.4$ | $19.4 \pm 1.8$ | $14.6 \pm 1.4$ | $14.4 \pm 0.3$ |
| ER | $19.0 \pm 0.6$ | $26.2 \pm 0.2$ | $20.0 \pm 0.8$ | $23.0 \pm 0.6$ | $24.0 \pm 2.0$ | $27.8 \pm 0.2$ | $18.7 \pm 1.6$ | $19.2 \pm 0.3$ |
| ER-RAR | $27.8 \pm 0.5$ | $36.2 \pm 0.7$ | $30.0 \pm 0.9$ | $36.5 \pm 0.4$ | $39.3 \pm 1.4$ | $45.0 \pm 2.7$ | $28.6 \pm 2.7$ | $28.9 \pm 1.5$ |
| GAINS | $8.8 \uparrow$ | $10.0 \uparrow$ | $10.0\uparrow$ | $13.5 \uparrow$ | $15.3 \uparrow$ | $17.2 \uparrow$ | $9.9 \uparrow$ | $9.7 \uparrow$ |
| MIR | $18.4 \pm 0.8$ | $25.7 \pm 1.8$ | $19.4 \pm 0.6$ | $22.3 \pm 0.2$ | $25.2 \pm 1.3$ | $26.9 \pm 0.9$ | $14.3 \pm 3.6$ | $15.2 \pm 3.0$ |
| MIR-RAR | $27.5 \pm 0.2$ | $36.1 \pm 0.3$ | $29.5 \pm 0.6$ | $34.9 \pm 0.7$ | $39.1 \pm 1.0$ | $44.6 \pm 1.7$ | $27.8 \pm 1.6$ | $29.2 \pm 2.6$ |
| GAINS | $9.1 \uparrow$ | $10.4 \uparrow$ | $10.1 \uparrow$ | $12.6 \uparrow$ | $13.9 \uparrow$ | $17.7 \uparrow$ | $13.5 \uparrow$ | $14.0 \uparrow$ |
| ASER | $20.9 \pm 0.3$ | $24.3 \pm 2.0$ | $15.7 \pm 0.1$ | $17.5 \pm 0.7$ | $16.4 \pm 1.4$ | $16.7 \pm 2.3$ | $19.4 \pm 1.3$ | $19.7 \pm 1.4$ |
| ASER-RAR | $28.1 \pm 0.3$ | $35.8 \pm 1.0$ | $27.0 \pm 0.3$ | $32.2 \pm 0.6$ | $24.2 \pm 0.4$ | $30.0 \pm 1.6$ | $28.7 \pm 0.2$ | $29.5 \pm 0.2$ |
| GAINS | $7.2 \uparrow$ | $11.5 \uparrow$ | $11.3 \uparrow$ | $14.7 \uparrow$ | $7.8 \uparrow$ | $13.3 \uparrow$ | $9.3 \uparrow$ | $9.8 \uparrow$ |
| SCR | $32.0 \pm 1.1$ | $37.4 \pm 0.2$ | $29.7 \pm 1.0$ | $33.1 \pm 1.9$ | $45.1 \pm 0.1$ | $50.3 \pm 1.9$ | $23.5 \pm 2.2$ | $23.6 \pm 3.0$ |
| SCR-RAR | $37.1 \pm 0.7$ | $45.8 \pm 0.2$ | $35.4 \pm 0.7$ | $43.7 \pm 0.4$ | $53.4 \pm 0.9$ | $61.1 \pm 1.1$ | $37.4 \pm 1.0$ | $41.5 \pm 0.9$ |
| GAINS | $5.1 \uparrow$ | $8.4 \uparrow$ | $5.7 \uparrow$ | $10.6 \uparrow$ | $8.3 \uparrow$ | $10.8 \uparrow$ | $14.9 \uparrow$ | $17.9 \uparrow$ |
| $ER_{RW}$ | $21.0 \pm 1.0$ | $26.2 \pm 0.2$ | $20.1 \pm 0.8$ | $23.0 \pm 0.6$ | $24.6 \pm 0.6$ | $27.8 \pm 0.8$ | $19.2 \pm 0.6$ | $19.2 \pm 0.3$ |
| $ER_{RW}$-RAR | $30.8 \pm 0.1$ | $36.5 \pm 0.4$ | $30.4 \pm 1.3$ | $36.5 \pm 0.4$ | $45.3 \pm 2.2$ | $50.8 \pm 0.9$ | $28.6 \pm 2.7$ | $28.9 \pm 1.5$ |
| GAINS | $9.8 \uparrow$ | $10.3 \uparrow$ | $10.3 \uparrow$ | $13.5 \uparrow$ | $20.7 \uparrow$ | $23.0 \uparrow$ | $9.4 \uparrow$ | $9.7 \uparrow$ |
| DER | $8.4 \pm 0.6$ | $9.1 \pm 0.3$ | $11.8 \pm 0.5$ | $12.3 \pm 1.7$ | $23.8 \pm 0.6$ | $23.4 \pm 2.5$ | $11.8 \pm 2.6$ | $12.6 \pm 1.1$ |
| DER-RAR | $30.0 \pm 1.2$ | $41.9 \pm 0.5$ | $26.2 \pm 0.4$ | $35.5 \pm 1.5$ | $37.7 \pm 1.4$ | $42.0 \pm 3.7$ | $28.4 \pm 3.2$ | $27.4 \pm 3.8$ |
| GAINS | $21.6 \uparrow$ | $32.8 \uparrow$ | $14.4 \uparrow$ | $23.2 \uparrow$ | $11.9 \uparrow$ | $18.6 \uparrow$ | $16.6\uparrow$ | $14.8\uparrow$ |

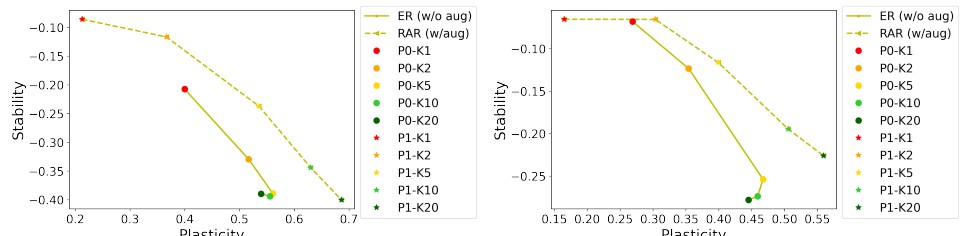

Figure 3: Stability and plasticity trade-off: CIFAR100 (left) and Mini-ImageNet(right)

**Implementation**: We use a reduced ResNet-18 for all datasets following Mai et al. [2021], Aljundi et al. [2019]. Single-head evaluation is employed with a shared final layer trained for all the tasks. RandAugmentation Cubuk et al. [2020] is used for auto augmentation. Given a set of augmentation operations, it randomly selects $P$ augmentation operations and exerts an augmentation magnitude of $Q$ for all the selected augmentation operations on each image. All the experimental results we present are averages of three runs. We summarize all hyperparameter details in Appendix C. The running time of different algorithms is shown in Appendix D.5.

## 6.2 Main Results

**RAR with ER and its variants** We first analyze RAR's performance with a pre-defined hyperparameter set ($K = 10, P = 1, Q = 14$). As shown in Table 1, RAR greatly improves the ER method on the four datasets, by $+8.8\% \sim +17.2\%$. Moreover, RAR also leads to substantial gains for the other algorithmic variants of ER for all datasets (MIR: $+9.1\% \sim +17.7\%$, ASER: $+7.2\% \sim +14.7\%$, SCR: $+5.1\% \sim +17.9\%$). These results suggest that even with advanced memory management strategies, such as MIR or ASER, or representation learning techniques, e.g., SCR, OCL still benefits substantially from repeated augmented rehearsal.

**RAR with Modified Rehearsal Loss** Besides using the vanilla online rehearsal loss in Eq 3, we investigate the effectiveness of RAR with another two, more advanced, rehearsal loss designs: 1) **Reweighted memory loss**: ER-rw introduces a reweighting hyperparameter $\alpha$ in the gradient of Eq 2 to deal with the biased ER loss by balancing the weight of the memory loss and incoming loss; 2) **Distillation-based memory loss**: DER [Buzzega et al., 2020] employs the logits-based distillation loss for memory samples, instead of the cross entropy loss. The results in Table 1 show

Table 2: Accuracy of variants of RAR and different hyperparameter tuning methods.

| | SEQ-CIFAR100 | SEQ-MINI-IMAGENET | CORE50-NC | CLRS25-NC |
|---|---|---|---|---|
| ER | $19.0 \pm 0.6$ | $20.0 \pm 0.8$ | $24.0 \pm 2.0$ | $18.7 \pm 1.6$ |
| RAR-MEM | $25.4 \pm 0.7$ | $27.4 \pm 0.8$ | $38.6 \pm 0.7$ | $28.8 \pm 1.0$ |
| RAR-INC | $21.6 \pm 0.2$ | $24.5 \pm 0.1$ | $35.7 \pm 1.1$ | $29.1 \pm 1.1$ |
| RAR-BOTH | $27.8 \pm 0.5$ | $30.0 \pm 0.9$ | $39.3 \pm 1.4$ | $28.6 \pm 2.7$ |
| RAR-HTOCL | $23.4 \pm 0.2$ | $26.0 \pm 0.2$ | $40.8 \pm 0.7$ | $26.9 \pm 0.5$ |
| RAR-RL | $\mathbf{29.6} \pm \mathbf{0.4}$ | $\mathbf{32.1} \pm \mathbf{1.0}$ | $\mathbf{44.4} \pm \mathbf{0.8}$ | $\mathbf{35.0} \pm \mathbf{0.7}$ |

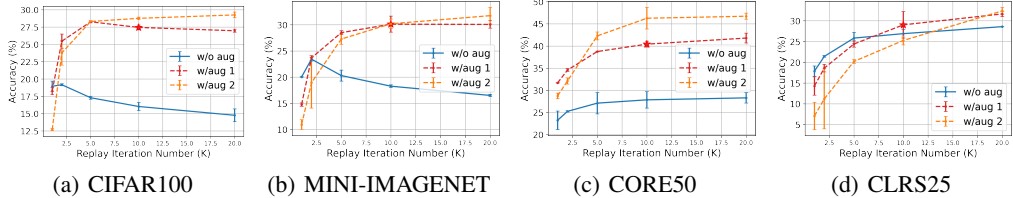

(a) CIFAR100  (b) MINI-IMAGENET  (c) CORE50  (d) CLRS25

Figure 4: Effects of augmentation and rehearsal iterations (red stars: accuracy in Table 1).

RAR leads to large performance gains for ER-rw (for the best $\alpha$ choice; further results can be found in Appendix D.2) and DER for all four datasets (ER-rw:$+9.4\% \sim +23.0\%$, DER:$+11.9\% \sim +32.8\%$). This suggests that even with more advanced rehearsal loss designs, repeated augmented rehearsal is important for online rehearsal.

**Stability and Plasticity Trade-off** Based on the definition of accuracy $A_T$, forgetting $F_T$ and backward transfer $B_T$ in Section 2, we find that these three metrics have the following relationship:

$$A_T = \underbrace{\frac{1}{T}\Sigma_{i=1}^{T}a_{i,i}}_{\text{Plasticity}} + \underbrace{\frac{T-1}{T}B_T}_{\text{Stability}} \geq \frac{1}{T}\Sigma_{i=1}^{T}a_{i,i} - \frac{T-1}{T}F_T.$$

Interestingly, this finding shows that accuracy is related to the ability to learn new tasks, quantified by $\frac{1}{T}\sum_{i=1}^{T}a_{i,i}$, and the ability to avoid forgetting past tasks, quantified by $\frac{T-1}{T}B_T$, which draws a connection to the more general problem of the stability-plasticity trade-off in neural networks and continual learning [Grossberg, 2012, Delange et al., 2021]. Plasticity refers to the ability to integrate new knowledge and stability refers to the ability to retain old knowledge. Fig 3 presents the stability and plasticity trade-off in RAR. Generally, we observe increasing the repeated rehearsal iterations ($K$) leads to a higher level of plasticity. However, this may also cause a decrease of stability, i.e., introduce forgetting. On the other hand, the use of augmentation generally improves stability. More importantly, the use of augmentation in repeated rehearsal shifts the stability-plasticity trade-off curve towards the upper right, thus creating a better stability-plasticity trade-off frontier.

**Hyperparameter Tuning for RAR** We compare the RL-based hyperparameter tuning method with the hyperparameter tuning framework for continual learning (HTOCL) method used in Chaudhry et al. [2018], Mai et al. [2022] (see Section 2). The results in Table 2 show that RL-based RAR significantly outperforms using HTOCL to select hyperparameters for RAR. One reason is that the extent of memory overfitting varies at the different training stages of CL. HTOCL only uses the first few tasks to select hyperparameters for RAR, which may not optimal for later stages of CL. In fact, we observe HTOCL tends to select a large number of repeats and small augmentation strength. This selection strategy may be desirable for problems with short task sequences but problematic for long task sequences with increased memory overfitting risk. In contrast, the RL-based method can take into account the latest feedback (e.g., train memory accuracy) to adjust hyperparameter choices. The selected iteration numbers and augmentation are shown in Appendix D.4.

## 6.3 Ablation Studies

**Interplay between Repeated and Augmented Rehearsal** We investigate the interaction of the number of replay iterations with the augmentation strength in Fig 4. For three out of four datasets, using augmentation alone without repeated rehearsal leads to even worse performances than rehearsal

without augmentation (see $K = 1$ in Fig 4). One explanation is the underfitting challenge of online CL. Training on augmented samples can make model underfitting even worse. The only exception is the CORE50 dataset, which inherently has a high memory overfitting risk with $\lambda = 6$ making it benefit more from augmentation. Similarly, employing repeated rehearsal alone (see the blue solid line in Fig 4) also harms performance in three out of four datasets, with the CLRS dataset as the only exception, which enjoys a low memory overfitting risk with $\lambda = 1.12$. An important takeaway message is that repeated rehearsal or augmented rehearsal is not always helpful in OCL settings and whether they will benefit or harm the performance is dependent on the structure of the OCL problem at hand (e.g., the task data size and memory data size).

**RAR's Robustness to Large Numbers of Repeats** Although the performance curve flattens out around 10 iterations, there is no evident drop in performance even with 20 iterations. This result reinforces how RAR can help with the underfitting-overfitting dilemma: it can support the use of more training in OCL without having to worry about the performance drop introduced by memory overfitting. In comparison, for repeated rehearsal without augmentation (solid lines), the accuracy starts to drop quickly on CIFAR100 and MiniImageNet when using more than two iterations.

**Augmenting the Memory vs. Augmenting the Incoming Data** RAR applies augmentation to both the memory batch and the incoming batch. We examine the effectiveness of the two separately. Table 2 presents the performance of RAR applied (a) solely with memory augmentation (RAR-mem) and (b) solely with incoming data augmentation (RAR-inc). We find that RAR achieves the best performance compared to RAR-mem and RAR-inc. This shows that it is beneficial to apply augmentation to both the memory batch and the incoming batch. Interestingly, RAR-inc itself also achieves consistent performance improvements over RER, e.g., a gain of 7.7% in CORE50. As discussed in Section 3, adding augmentation on the incoming batch can strengthen the regularization effect of the incoming task and indirectly alleviate memory overfitting.

**RAR with MIR, ASER, SCR** We also perform ablation studies to investigate RAR's strong performance with the ER variants (see Appendix D.1). Similar to ER, the performance gains for RAR-MIR and RAR-ASER come from the combination of repeated rehearsal and augmented rehearsal. However, for SCR, the repeated rehearsal itself also leads to a consistent performance boost. One reason is that SCR already includes a strong augmentation procedure with four augmentation operations to construct the supervised contrastive loss used in this method. It also works poorly without augmentation.

**Augmentation for Offline Rehearsal** Although this paper focuses on online CL, the (augmented) empirical risk analysis is also relevant to offline CL (see Propositions 2 and 3 in Appendix A). This suggests the memory overfitting risk in offline rehearsal is related to the ratio of task-to-memory size $\lambda$ and can be alleviated by augmented rehearsal (see more empirical results in Table 8 of Appendix D.6).

# 7   Discussion and Conclusion

Rehearsal-based methods play a central role in fighting catastrophic forgetting when learning from non-stationary data streams. Compared to offline rehearsal, online rehearsal has faced particular challenges in tackling complex CL datasets due to the single-pass-through data constraint. This work tries to analyze the internal workings of online rehearsal from a theoretical and conceptual perspective and identifies the fundamental challenge that it faces as the dilemma between overfitting locally and underfitting globally. To deal with this challenge, we propose a simple baseline: repeated and augmented rehearsal (RAR). Surprisingly, despite its simplicity, RAR achieves a large performance boost for a set of different rehearsal-based methods. Additionally, we propose an RL-based method to tune the hyperparameters of RAR to balance the stability-plasticity trade-off in an online manner. It achieves promising results compared to hyperparameter tuning based on validation data. This work is focused on continual learning for classification problems with image data. An interesting future research direction is to look at other CL domains, e.g., text/audio inputs or RL problems.

## Acknowledgments and Disclosure of Funding

This research is funded by the New Zealand MBIE TAIAO data science programme.

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
