# A Theoretical Analysis

This section contains the theoretical analysis of the loss functions of offline experience replay (**Proposition 2**), augmented experience replay (**Proposition 3**), and online experience replay with reservoir sampling (**Proposition 1**).

**Proposition 2 (Empirical risk minimization for experience replay)**: We assume a memory set $\mathcal{D}_\mathcal{M}$ and an incoming task stream $\mathcal{D}_\mathcal{T}$ with different data distribution $\mathbb{P}(\mathcal{D}_\mathcal{T}) \neq \mathbb{P}(\mathcal{D}_\mathcal{M})$. At each iteration $t$, a batch of data is sampled from memory $\mathcal{B}_t^\mathcal{M} \sim \mathcal{D}_\mathcal{M}$ with $\mathcal{B}_t^\mathcal{M} = \{\mathbf{x}_i, y_i\}_{i=1,\dots|\mathcal{B}_t^\mathcal{M}|}$, and a batch of data is sampled from incoming task $\mathcal{B}_t \sim \mathcal{D}_\mathcal{T}$ with $\mathcal{B}_t = \{\mathbf{x}_i, y_i\}_{i=1,\dots|\mathcal{B}_t|}$. To update the parameter $\theta$ of a function $f_\theta$: $f_\theta \times \mathcal{X} \to \mathcal{Y}$ based on loss function $\mathcal{L}$, consider the parameter update rule defined by $\theta = \theta - \frac{\eta}{|\mathcal{B}_t|} \sum_{\mathbf{x}_i, y_i \in \mathcal{B}_t} \nabla \mathcal{L}(f_\theta(\mathbf{x}_i), y_i) - \frac{\eta}{|\mathcal{B}_t^\mathcal{M}|} \sum_{\mathbf{x}_i, y_i \in \mathcal{B}_t^\mathcal{M}} \nabla \mathcal{L}(f_\theta(\mathbf{x}_i), y_i)$, then it is an unbiased stochastic gradient descent update rule for the following empirical risk:

$$\mathcal{R}(\theta) = \sum_{x_i, y_i \in \mathcal{D}_\mathcal{T}} \mathcal{L}(f_\theta(\mathbf{x}_i), y_i) + \frac{|\mathcal{D}_\mathcal{T}|}{|\mathcal{D}_\mathcal{M}|} \sum_{x_i, y_i \in \mathcal{D}_\mathcal{M}} \mathcal{L}(f_\theta(\mathbf{x}_i), y_i).$$

**Proof**: Given the empirical gradient $\hat{g}_{ER} = \frac{1}{|\mathcal{B}_t|} \sum_{\mathbf{x}_i, y_i \in \mathcal{B}_t} \nabla \mathcal{L}(f_\theta(\mathbf{x}_i), y_i) + \frac{1}{|\mathcal{B}_t^\mathcal{M}|} \sum_{\mathbf{x}_i, y_i \in \mathcal{B}_t^\mathcal{M}} \nabla \mathcal{L}(f_\theta(\mathbf{x}_i), y_i)$ during stochastic optimization, we can derive the gradient expectation as follows:

$$\mathbb{E}_{\mathcal{B}_t^\mathcal{M} \sim \mathcal{D}_\mathcal{M}, \mathcal{B}_t \sim \mathcal{D}_\mathcal{T}}[\hat{g}_{ER}] = \nabla(\mathbb{E}_{\mathcal{B}_t^\mathcal{M} \sim \mathcal{D}_\mathcal{M}}[\frac{1}{|\mathcal{B}_t^\mathcal{M}|} \sum_{\mathcal{B}_t^\mathcal{M}} \mathcal{L}(f_\theta(\mathbf{x}_i), y_i)] + \mathbb{E}_{\mathcal{B}_t \sim \mathcal{D}_\mathcal{T}}[\frac{1}{|\mathcal{B}_t|} \sum_{\mathcal{B}_t} \mathcal{L}(f_\theta(\mathbf{x}_i), y_i)])$$

$$= \nabla(\mathbb{E}_{x_i, y_i \sim \mathcal{D}_\mathcal{M}}[\mathcal{L}(f_\theta(\mathbf{x}_i), y_i)] + \mathbb{E}_{x_i, y_i \sim \mathcal{D}_\mathcal{T}}[\mathcal{L}(f_\theta(\mathbf{x}_i), y_i)])$$

$$= \nabla(\frac{1}{|\mathcal{D}^\mathcal{M}|} \sum_{\mathcal{D}_\mathcal{M}} \mathcal{L}(f_\theta(\mathbf{x}_i), y_i) + \frac{1}{|\mathcal{D}_\mathcal{T}|} \sum_{\mathcal{D}_\mathcal{T}} \mathcal{L}(f_\theta(\mathbf{x}_i), y_i))$$

$$= \frac{1}{|\mathcal{D}_\mathcal{T}|} \nabla \left( \frac{|\mathcal{D}_\mathcal{T}|}{|\mathcal{D}_\mathcal{M}|} \sum_{\mathcal{D}_\mathcal{M}} \mathcal{L}(f_\theta(\mathbf{x}_i), y_i) + \sum_{\mathcal{D}_\mathcal{T}} \mathcal{L}(f_\theta(\mathbf{x}_i), y_i) \right)$$

**Proposition 3 (Augmented empirical risk minimization for experience replay)**: Assume a memory set $\mathcal{D}_\mathcal{M} = \{\mathbf{x}_i, y_i\}_{i=1,\dots,|\mathcal{D}_\mathcal{M}|}$ and an incoming task stream $\mathcal{D}_\mathcal{T} = \{\mathbf{x}_i, y_i\}_{i=1,\dots,|\mathcal{D}_\mathcal{T}|}$ with different data distribution $\mathbb{P}(\mathcal{D}_\mathcal{T}) \neq \mathbb{P}(\mathcal{D}_\mathcal{M})$, and a compact topological group of transform $G$ with a probability distribution $\mathbb{Q}$ that acts on the input space $\mathcal{X}$ and is invariant under function $f$, i.e., $f(gx) = f(x), g \in G, x \in \mathcal{X}$. At each iteration $t$, a batch of data is sampled from memory, $\mathcal{B}_t^\mathcal{M} \sim \mathcal{D}_\mathcal{M}$ and $\mathcal{B}_t^\mathcal{M} = \{\mathbf{x}_i, y_i\}_{i=1,\dots|\mathcal{B}_t^\mathcal{M}|}$, a batch of data is sampled from the incoming task, $\mathcal{B}_t \sim \mathcal{D}_\mathcal{T}$ and $\mathcal{B}_t = \{\mathbf{x}_i, y_i\}_{i=1,\dots|\mathcal{B}_t|}$, and a group operation $g_t \sim G$ is randomly selected. To update the parameter $\theta$ of the function $f_\theta$ based on loss function $\mathcal{L}$, consider the parameter update rule defined by $\theta = \theta - \frac{\eta}{|\mathcal{B}_t|} \sum_{\mathbf{x}_i, y_i \in \mathcal{B}_t} \nabla \mathcal{L}(f_\theta(g_t\mathbf{x}_i), y_i) - \frac{\eta}{|\mathcal{B}_t^\mathcal{M}|} \sum_{\mathbf{x}_i, y_i \in \mathcal{B}_t^\mathcal{M}} \nabla \mathcal{L}(f_\theta(g_t\mathbf{x}_i), y_i)$, then it is an unbiased stochastic gradient descent update rule for the loss function

$$\bar{\mathcal{R}}(\theta) = \sum_{x_i, y_i \in \mathcal{D}_\mathcal{T}} \int_G \mathcal{L}(f_\theta(gx_i), y_i) d\mathbb{Q}(g) + \frac{|\mathcal{D}_\mathcal{T}|}{|\mathcal{D}_\mathcal{M}|} \sum_{x_i, y_i \in \mathcal{D}_\mathcal{M}} \int_G \mathcal{L}(f_\theta(gx_i), y_i) d\mathbb{Q}(g).$$

**Proof**:

Given the augmented empirical gradient during stochastic optimization, we can derive the gradient expectation as follows:

$$\mathbb{E}_{\mathcal{B}_t^\mathcal{M} \sim \mathcal{D}_\mathcal{M}, \mathcal{B}_t \sim \mathcal{D}_\mathcal{T}, g \sim \mathbb{Q}}[\hat{g}]$$

$$= \nabla(\mathbb{E}_{\mathcal{B}_t^\mathcal{M} \sim \mathcal{D}_\mathcal{M}}[\frac{1}{|\mathcal{B}_t^\mathcal{M}|} \sum_{\mathcal{B}_t^\mathcal{M}} \int_g \mathcal{L}(f_\theta(g\mathbf{x}_i), y_i) d\mathbb{Q}(g)] + \mathbb{E}_{\mathcal{B}_t \sim \mathcal{D}_\mathcal{T}}[\frac{1}{|\mathcal{B}_t|} \sum_{\mathcal{B}_t} \int_g \mathcal{L}(f_\theta(\mathbf{x}_i), y_i) d\mathbb{Q}(g)])$$

$$= \frac{1}{|\mathcal{D}_\mathcal{T}|} \nabla \left( \sum_{x_i, y_i \in \mathcal{D}_\mathcal{T}} \int_G \mathcal{L}(f_\theta(gx_i), y_i) d\mathbb{Q}(g) + \frac{|\mathcal{D}_\mathcal{T}|}{|\mathcal{D}_\mathcal{M}|} \sum_{x_i, y_i \in \mathcal{D}_\mathcal{M}} \int_G \mathcal{L}(f_\theta(gx_i), y_i) d\mathbb{Q}(g) \right)$$

The second equality is based on the results of Proposition 1, which we prove next.

**Proposition 1 (ERM for online rehearsal)**: Assume an initial memory set $\mathcal{D}_{\mathcal{M}}^0$ and an incoming task stream $\mathcal{D}_{\mathcal{T}}$ with different data distribution $\mathbb{P}(\mathcal{D}_{\mathcal{T}}) \neq \mathbb{P}(\mathcal{D}_{\mathcal{M}})$. At each iteration $t$, $t = 1, ..T$, a batch of data is sampled from the incoming task, $\mathcal{B}_t \sim \mathcal{D}_{\mathcal{T}}$ and $\mathcal{B}_t = \{\mathbf{x}_i, y_i\}_{i=1,...|\mathcal{B}_t|}$, and a batch of data is sampled from the memory, $\mathcal{B}_t^{\mathcal{M}} \sim \mathcal{D}_{\mathcal{M}}^t$ and $\mathcal{B}_t^{\mathcal{M}} = \{\mathbf{x}_i, y_i\}_{i=1,...|\mathcal{B}_t^{\mathcal{M}}|}$. To update the parameter $\theta$ of the function $f_\theta$: $f_\theta \times \mathcal{X} \to \mathcal{Y}$ based on loss function $\mathcal{L}$, consider a parameter update rule defined by $\theta = \theta - \frac{\eta}{|\mathcal{B}_t|} \sum_{\mathbf{x}_i, y_i \in \mathcal{B}_t} \nabla \mathcal{L}\left(f_\theta(\mathbf{x}_i), y_i\right) - \frac{\eta}{|\mathcal{B}_t^{\mathcal{M}}|} \sum_{\mathbf{x}_i, y_i \in \mathcal{B}_t^{\mathcal{M}}} \nabla \mathcal{L}\left(f_\theta(\mathbf{x}_i), y_i\right)$. Assume the memory is updated at the end of each iteration using reservoir sampling Vitter [1985], Chaudhry et al. [2019] $\mathcal{D}_{\mathcal{M}}^{t+1} \leftarrow RS(\mathcal{D}_{\mathcal{M}}^t, \mathcal{B}_t)$. Then the gradient update rule is an unbiased stochastic gradient descent update rule for the loss function

$$\mathcal{R}_t(\theta) = \sum_{x_i, y_i \in \mathcal{D}_{\mathcal{T}}} \mathcal{L}(f_\theta(x_i), y_i) + \beta_t \frac{|\mathcal{D}_{\mathcal{T}}|}{|\mathcal{D}_{\mathcal{M}}^0|} \sum_{x_i, y_i \in \mathcal{D}_{\mathcal{M}}^0} \mathcal{L}(f_\theta(x_i), y_i)$$

where $\beta_t = \frac{1}{1+2*\frac{N_{cur}^t}{N_{past}}}$, $N_{cur}^t = \sum_{i=1}^{i=t} |\mathcal{B}_i|$ denotes the number of samples of the current task that have been seen so far and $N_{past} = \sum_{j=1}^{j=\mathcal{T}} |\mathcal{D}_j|$ denotes the number of samples of past tasks.

**Note 1**: The objective function $\mathcal{R}_t(\theta)$ changes with respect to the batch number $t$ due to the changes in $\beta_t$.

**Note 2**: $\frac{1}{1+2*\frac{|\mathcal{D}_{\mathcal{T}}|}{N_{past}}} \leq \beta_t \leq 1$ and $\beta_t$ is decreasing with the batch number $t$. At the start of a task, $\beta_{t=0} = \beta_{max} = 1$. At the end of a task, $\beta_{t=T} = \beta_{min} = \frac{1}{1+2*\frac{|\mathcal{D}_{\mathcal{T}}|}{N_{past}}}$.

**Note 3**: Consider a balanced continual learning dataset (e.g., Split-CIFAR100, Split-Mini-ImageNet) where $|\mathcal{D}_j| = |\mathcal{D}_{\mathcal{T}}|, j = 1, ..\mathcal{T}$. Then, we have $\beta_{min} = \frac{\mathcal{T}-1}{\mathcal{T}+1} = (1 - \frac{2}{\mathcal{T}+1})$ and $\lim_{\mathcal{T} \to \infty} \beta_{min} = 1$.

**Note 4**: Consider general continual learning datasets. As CL learns more tasks, $N_{past}$ increases, and $\lim_{N_{past} \to \infty} \beta_t = \lim_{N_{past} \to \infty} \frac{1}{1+2*\frac{N_{cur}^t}{N_{past}}} = 1$.

**Proof**: Given the empirical gradient $\hat{g}_{ER} = \frac{1}{|\mathcal{B}_t|} \sum_{\mathbf{x}_i, y_i \in \mathcal{B}_t} \nabla \mathcal{L}\left(f_\theta(\mathbf{x}_i), y_i\right) + \frac{1}{|\mathcal{B}_t^{\mathcal{M}}|} \sum_{\mathbf{x}_i, y_i \in \mathcal{B}_t^{\mathcal{M}}} \nabla \mathcal{L}\left(f_\theta(\mathbf{x}_i), y_i\right)$ during stochastic optimization, we can derive the gradient expectation as follows:

$$\mathbb{E}_{\mathcal{D}_{\mathcal{M}}^t}[\mathbb{E}_{\mathcal{B}_t^{\mathcal{M}} \sim \mathcal{D}_{\mathcal{M}}^t, \mathcal{B}_t \sim \mathcal{D}_{\mathcal{T}}}[\hat{g}_{ER}]]$$

$$= \nabla \mathbb{E}_{\mathcal{D}_{\mathcal{M}}^t} \left[ \mathbb{E}_{\mathcal{B}_t^{\mathcal{M}} \sim \mathcal{D}_{\mathcal{M}}^t}[\frac{1}{|\mathcal{B}_t^{\mathcal{M}}|} \sum_{\mathcal{B}_t^{\mathcal{M}}} \mathcal{L}\left(f_\theta(\mathbf{x}_i), y_i\right)] + \mathbb{E}_{\mathcal{B}_t \sim \mathcal{D}_{\mathcal{T}}}[\frac{1}{|\mathcal{B}_t|} \sum_{\mathcal{B}_t} \mathcal{L}\left(f_\theta(\mathbf{x}_i), y_i\right)] \right]$$

$$= \nabla (\mathbb{E}_{\mathcal{D}_{\mathcal{M}}^t} \left[ \mathbb{E}_{x_i, y_i \sim \mathcal{D}_{\mathcal{M}}^t}[\mathcal{L}\left(f_\theta(\mathbf{x}_i), y_i\right)] \right] + \mathbb{E}_{x_i, y_i \sim \mathcal{D}_{\mathcal{T}}}[\mathcal{L}\left(f_\theta(\mathbf{x}_i), y_i\right)])$$

$$= \nabla (\frac{N_{past}}{N_{past} + N_{cur}^t} \mathbb{E}_{x_i, y_i \sim \mathcal{D}_{\mathcal{M}}^0}[\mathcal{L}\left(f_\theta(\mathbf{x}_i), y_i\right)] + (\frac{N_{cur}^t}{N_{past} + N_{cur}^t} + 1)\mathbb{E}_{x_i, y_i \sim \mathcal{D}_{\mathcal{T}}}[\mathcal{L}\left(f_\theta(\mathbf{x}_i), y_i\right)])$$

$$= \frac{N_{past} + 2 * N_{cur}^t}{(N_{past} + N_{cur}^t) * |\mathcal{D}_{\mathcal{T}}|} \nabla \left( \frac{N_{past}}{N_{past} + 2 * N_{cur}^t} \frac{|\mathcal{D}_{\mathcal{T}}|}{|\mathcal{D}_{\mathcal{M}}^0|} \sum_{\mathcal{D}_{\mathcal{M}}^0} \mathcal{L}\left(f_\theta(\mathbf{x}_i), y_i\right) + \sum_{\mathcal{D}_{\mathcal{T}}} \mathcal{L}\left(f_\theta(\mathbf{x}_i), y_i\right) \right)$$

The third equality is based on the condition that $\mathcal{D}_{\mathcal{M}}^t$ is updated using reservoir sampling so that all the data seen so far have an equal probability of being stored in the memory. In other words, at a given time $t$, the memory contains a fraction of $\frac{N_{past}}{N_{past}+N_{cur}^t}$ samples coming from past tasks and a fraction of $\frac{N_{curr}^t}{N_{past}+N_{cur}^t}$ samples coming from the current task.

# B  Dataset details

Table 3 lists the image size, the number of classes, the number of tasks, and data size per task of the four CL benchmarks.

Table 3: Dataset information for the four CL benchmarks.

|  | IMAGE SIZE | #TASK | # CLASS | TRAIN PER TASK | TEST PER TASK |
|---|---|---|---|---|---|
| SEQ-CIFAR100 | 3x32x32 | 20 | 100 | 2,500 | 500 |
| SEQ-MINI-IMAGENET | 3x84x84 | 10 | 100 | 5,000 | 1,000 |
| CORE50-NC | 3x128x128 | 9 | 50 | 12,000-24,000 | 4,500-9,000 |
| CLRS-NC | 3x256x256 | 5 | 25 | 2,250 | 750 |

## C  Implementation Details

### C.1  Continual Learning Implementation

The hyperparameter settings are summarized in Table 4. All models are optimized using vanilla SGD. For all experiments, we use the learning rate of 0.1 following the same setting as in Aljundi et al. [2019], Shim et al. [2021], and the Nearest-Class-Mean (NCM) classifier is used for evaluation, as Mai et al. reported (2021) considerable and consistent performance gains when replacing the Softmax classifier with the NCM classifier. Each mini batch during training consists of 10 new and 10 memory samples, except for the SCR method, which employs 100 memory samples and 10 new incoming samples Mai et al. [2021]. By default, the repeated rehearsal parameter for all the results is $K = 10$ and the augmented rehearsal parameters are $P = 1, Q = 14$.

This paper uses Randaugment [Cubuk et al., 2020], which is an auto augmentation method. It randomly selects $P$ augmentation operators from a set of 14 operators and applies them to the images. The augmentation operator set includes:'Identity', 'AutoContrast', 'Equalize', 'Rotate', 'Solarize', 'Color','Posterize', 'Contrast', 'Brightness', 'Sharpness','ShearX', 'ShearY','TranslateX', 'TranslateY'.

Table 4: Hyperparameter setting.

|  | HYPERPARAMETER |
|---|---|
| LWF | LR=0.1 |
| AGEM | LR=0.1 |
| ER | BATCHSIZE=10,LR=0.1 |
| MIR | BATCHSIZE=10,C=50,LR=0.1 |
| ASER | K=3, N_SMP_CLS=1.5,BATCHSIZE=10,LR=0.1 |
| SCR | TEMP =0.07, BATCH SIZE = 100,LR=0.1 |
| DER | $\alpha = 0.3$, AUGMENTATION: FLIP AND CROP, $K = 50$ |

### C.2  RL-based hyperparameter tuning implementation

The memory iteration choices are from 1 to 20 and the augmentation choices are $(1, 5), (1, 14), (2, 14), (3, 14), (4, 14)$. Action selection probabilities are modeled with softmax weight $\pi_w(a_i) = \frac{e^{w_i}}{\sum_k e^{w_k}}$. Bootstrapped policy gradient is used to adjust action weights:

$$g^{BPG} = \mathbb{E}_{a_i \sim \pi_w} \left[ |r_{a_i}| \left( \nabla_w \log \widehat{\pi}_w^+ (a_i) - \nabla_w \log \widehat{\pi}_w^- (a_i) \right) \right]$$

where $\hat{\pi}_w^+ (a_i) := \Sigma_{a_k \in \mathcal{X}_{a_i}^+} \pi_w (a_k)$ and $\hat{\pi}_w^- (a_i) := \Sigma_{a_k \in \mathcal{X}_{a_i}^-} \pi_w (a_k)$

**Better/Worse action set** The key of the BPG idea is to incorporate prior information into the construction of better and worse action sets. To apply BPG in the OCL environment, we propose to determine the better/worse action set based on the feedback in the form of current memory batch accuracy $A_{\mathcal{M}}$, which reflects the memory overfitting level of the CL agent. We want the training memory accuracy to be neither too high nor too low. The desirable training memory accuracy is defined by $A_{\mathcal{M}}^*$. We find a higher repeated replay iteration leads to higher memory accuracy.

Therefore, the better/worse action set is defined as follows:

$$
\mathcal{X}_a^+ := \begin{cases} \forall a_k \mid Iter\,(a_k) < Iter(a), & A_{\mathcal{M}}(a) > A_{\mathcal{M}}^* \\ \forall a_k \mid Iter\,(a_k) > Iter(a), & A_{\mathcal{M}}(a) < A_{\mathcal{M}}^* \\ \emptyset, & A_{\mathcal{M}}(a) = A_{\mathcal{M}}^* \end{cases} \quad \mathcal{X}_a^- := \begin{cases} \forall a_k \mid Iter\,(a_k) > Iter(a), & A_{\mathcal{M}}(a) > A_{\mathcal{M}}^* \\ \forall a_k \mid Iter\,(a_k) < Iter(a), & A_{\mathcal{M}}(a) < A_{\mathcal{M}}^* \\ \forall a_k \mid Iter\,(a_k) \neq Iter(a), & A_{\mathcal{M}}(a) = A_{\mathcal{M}}^* \end{cases}
$$

$$
\mathcal{X}_a^+ := \begin{cases} \forall a_k \mid Aug\,(a_k) > Aug(a), & A_{\mathcal{M}}(a) > A_{\mathcal{M}}^* \\ \forall a_k \mid Aug\,(a_k) < Aug(a), & A_{\mathcal{M}}(a) < A_{\mathcal{M}}^* \\ \emptyset, & A_{\mathcal{M}}(a) = A_{\mathcal{M}}^* \end{cases} \quad \mathcal{X}_a^- := \begin{cases} \forall a_k \mid Aug\,(a_k) < Aug(a), & A_{\mathcal{M}}(a) > A_{\mathcal{M}}^* \\ \forall a_k \mid Aug\,(a_k) > Aug(a), & A_{\mathcal{M}}(a) < A_{\mathcal{M}}^* \\ \forall a_k \mid Aug\,(a_k) \neq Aug(a), & A_{\mathcal{M}}(a) = A_{\mathcal{M}}^* \end{cases}
$$

**Reward** A challenge in the hyperparameter tuning for the OCL setting is that the CL agent may face new tasks with unseen data distribution. Therefore, it is often infeasible to assume the existence of an external validation data containing all the tasks in advance for hyperparameter tuning. To achieve online hyperparameter tuning without external validation data, we propose to define the reward based on the accuracy on the memory. Given a target memory accuracy $A_{\mathcal{M}}^*$, the reward is defined as $r = |A_{\mathcal{M}} - A_{\mathcal{M}}^*|$

**Non-stationarity** To address the non-stationary nature in the CL environment, we reset the weight of BPG to a uniform weight at the start of each task.

The analysis of the selected action can be found in Figures 9 and 10 in Section D.4.

Algorithm 2: BPG-based RAR

---

$\mathcal{M}$ is the memory with fixed size,
$\mathcal{B}_t$ is the incoming batch from the current task,
$\theta$ are the parameters of the CL network,
$w$ are the parameters of the RL agent,
$K$ is the number of memory iterations,
$P, Q$ are the augmentation hyperparameters
$A_{\mathcal{M}}^*$ target memory accuracy

1: **procedure** RAR($\mathcal{M}_t, \mathcal{B}_t, \theta_t, w_t$ )
2:      $K_t \sim \pi_{w_t^{iter}}$
3:      $P_t, Q_t \sim \pi_{w_t^{aug}}$
4:      **for** $k = 1, ..., K_t$ **do**
5:          $\mathcal{B}_{t,k}^{\mathcal{M}} \sim MemRetrieval(\mathcal{M}_t)$
6:          $\mathcal{B}_{aug} \leftarrow aug(\mathcal{B}_{t,k}^{\mathcal{M}} \cup \mathcal{B}_t, P_t, Q_t)$
7:          $A_{\mathcal{M}} \leftarrow MemAcc(\mathcal{M}, \theta_{t,k})$
8:          $\mathcal{X}^+, \mathcal{X}^- \leftarrow ActionSet(A_{\mathcal{M}}, A_{\mathcal{M}}^*)$
9:          $\theta_{t,k+1} \leftarrow SGD(\mathcal{B}_{aug}, \theta_{t,k})$
10:     **end for**
11:     $\mathcal{M}_{t+1} \leftarrow MemUpdate(\mathcal{M}_t, \mathcal{B}_t)$
12:     $w_{t+1}^{iter}, w_{t+1}^{aug} \leftarrow UpdateRL(A_{\mathcal{M}}, \mathcal{X}^+, \mathcal{X}^-)$
13: **end procedure**

---

# D  Supplementary Experimental Results

## D.1   Ablation studies for MIR-RAR, ASER-RAR and SCR-RAR

As shown in Table 1, RAR improves ER and ER variants (MIR, ASER and SCR). The detailed results of the ablation studies of RAR with ER variants are presented in this section. In particular, Fig 5 and 6 show the comparison of only using repeated rehearsal or augmented rehearsal for MIR and ASER respectively. Neither of them alone consistently improves the performance of the baseline. This result suggests the influence of RAR for MIR/ASER is similar to ER. The performance gain comes from the combination of the repeated rehearsal and augmented rehearsal. Fig 7 shows the results for SCR. Repeated rehearsal leads to consistent performance gains because SCR already includes augmentation.

## D.2   Reweighted ER

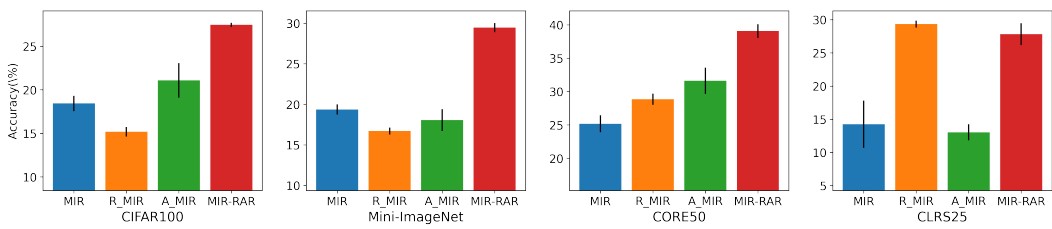

Figure 5: Ablation study: MIR-RAR on the four datasets, with a 2k memory

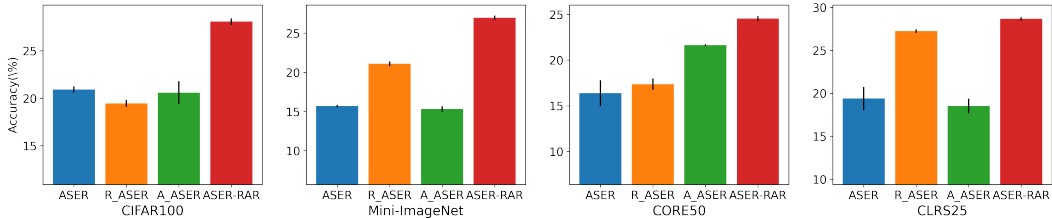

Figure 6: Ablation study: ASER-RAR on the four datasets, with a 2k memory

To deal with the biased ER loss, one straightforward way is to balance the weight of the memory loss and incoming loss by introducing a reweighting hyperparameter $\alpha$ in the gradient of Eq 2. Specifically, the gradient for reweighted ER is implemented as

$$g_t^{ER-rw} = (1 - \alpha)\frac{1}{|B_t|} \sum_{x,y \in B_t} \nabla L(f_\theta(x), y) + \alpha\frac{1}{|B_t^M|} \sum_{x,y \in B_t^M} \nabla L(f_\theta(x), y), \alpha \in (0, 1).$$

The performance of ER and RAR with respect to different reweighting hyperparameter values $\alpha \in [0.1, 0.3, 0.5, 0.7, 0.9]$ is shown in Fig 8, where $\alpha = 0.5$ denotes vanilla ER loss with the equal weighting of memory loss and incoming loss. To keep the learning rate comparable to vanilla ER, the learning rate of ER-rw is twice that of ER. A key observation is that similar to vanilla ER, reweighted ER (ER-rw) also significantly benefits from repeated and augmented rehearsal, as ER-rw-rar greatly improves over ER-rw (see the red line and blue line in Fig 8) for all four datasets.

### D.3 Large-scale online CL

To examine the effectiveness of RAR in a large-scale continual learning problem, we apply RAR to ImageNet-1k. ImageNet-1k is split into 10 tasks and each task contains 100 classes. The training dataset of 10 tasks contains 1,281,167 images in total. Considering the task size, we evaluate with a memory size of M= 20k (with the task to memory size ratio $\lambda \approx 6.4$) and a memory size of M=100k (with the task to memory size ratio $\lambda \approx 1.28$). These choices are similar to Seq-CIFAR100 ($\lambda = 1.25$) and CORE50 ($\lambda = 6$) with a 2k memory. We randomly crop the images to size 224x224, and use the ResNet-18 architecture for training on ImageNet-1k with a single epoch. An incoming batch size of 32 and a memory batch size of 32 are used. We employ a learning rate of 0.1 with 0.001 of weight decay. For the RAR method, we use 10 memory iterations with RandAugment and parameters P=1 and Q=14. The results are shown in the Table 5.

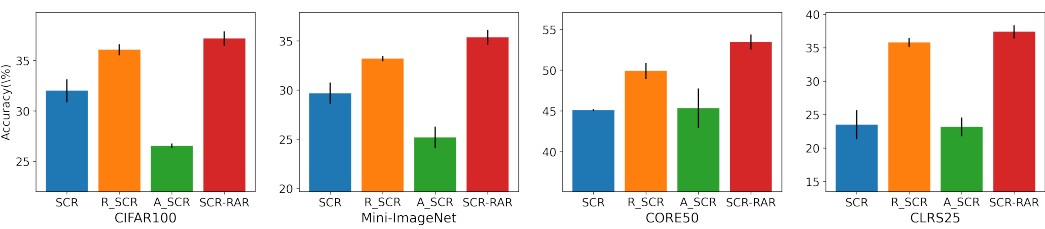

Figure 7: Ablation study: SCR-RAR on the four datasets, with a 2k memory

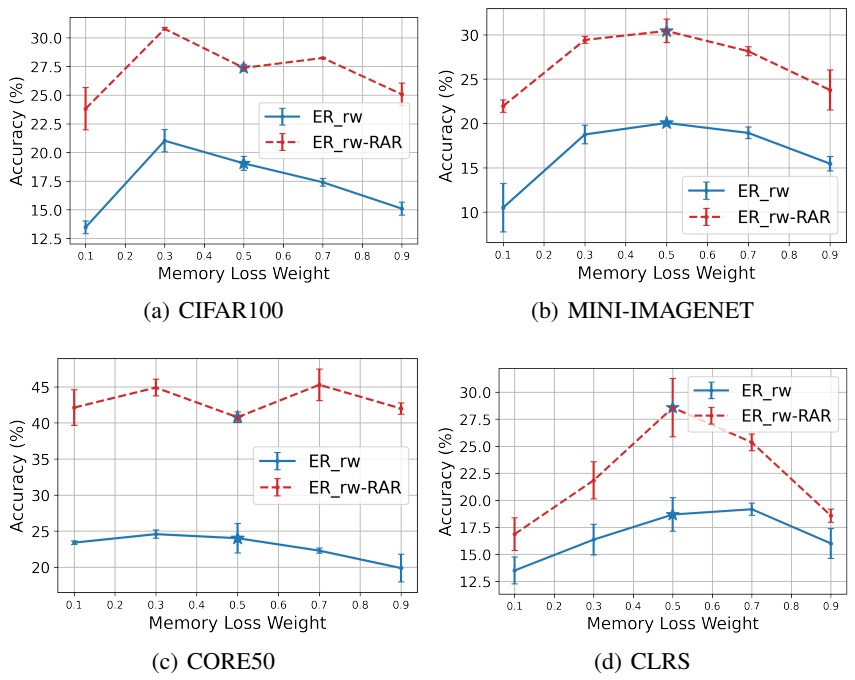

Figure 8: The performance of reweighting the memory loss of ER (ER-rw) and its effectiveness with RAR on the four datasets. (Star symbols denote the accuracy of ER and ER-RAR with a reweighting value of 0.5).

We observe that repeated augmented rehearsal (RAR) is effective in this large-scale online continual learning problem and improves the vanilla rehearsal from 2.1% to 15.1% with a 20k memory and from 8.7% to 34.7% with a 100k memory.

The discussion in Buzzega et al. [2020] suggests that to deal with complex continual learning datasets, it is necessary to employ multiple epochs of training with offline CL to avoid the underfitting problem present in online learning. Our results show that RAR can greatly improve online rehearsal for large-scale CL problems. Although several offline CL algorithms have been evaluated on ImageNet-1k, to our knowledge, our result is the first attempt to apply online CL to this problem. We aim to investigate other online rehearsal-based methods on ImageNet-1k in future work.

Table 5: Accuracy of ER and ER-RAR for ImageNet-1k with a 20k and 100k memory.

| IMAGENET-1K | M=20K | M=100K |
|---|---|---|
| ER | $2.1 \pm 0.3$ | $8.7 \pm 0.5$ |
| ER-RAR | $15.1 \pm 0.4$ | $34.7 \pm 0.1$ |
| GAINS | $13.0 \uparrow$ | $26.0 \uparrow$ |

### D.4 Action Selection in Reinforcement Learning

The selected hyperparameters for four datasets are shown in Figure 9. Interestingly, the RL-based method assigns a stronger augmentation and lower iteration for a dataset with a higher $\lambda$ attribute. As discussed in the ERM analysis, a dataset with a higher $\lambda$ suggests a higher risk of overfitting. The RL-based method successfully takes this into account to tune the hyperparameters. In contrast, the OCL-HT method selects the weakest augmentation and highest iteration for three datasets.

Figure 10 presents the selected hyperparameters at different CL training stages. Generally, as the continual learning proceeds, the RL-based method selects a stronger augmentation strength and lower iterations, to balance off the increasing risk of memory overfitting.

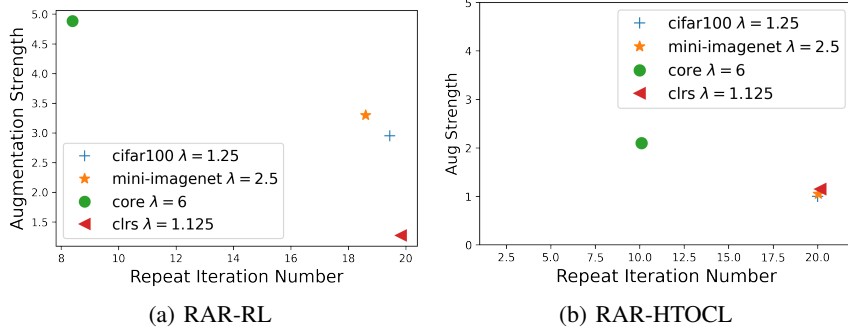

(a) RAR-RL          (b) RAR-HTOCL

Figure 9: The average of selected hyperparameters of RAR (iteration and augmentation values) for four datasets.

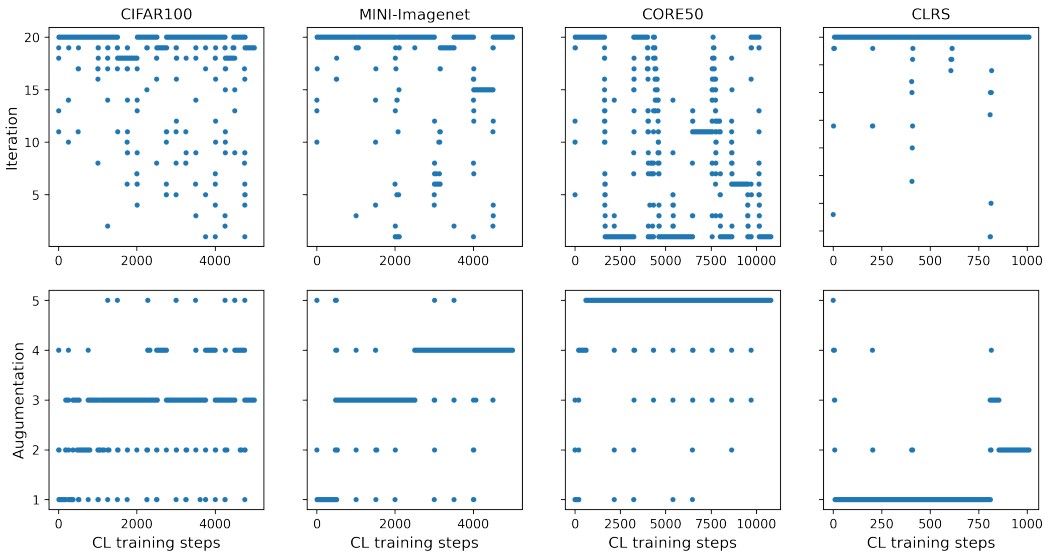

Figure 10: The selected hyperparameters of RAR (iteration and augmentation values) of the RL-based method.

## D.5 Running time

### D.5.1 Running time of RAR

The running time of ER and ER-RAR is shown in Table 6. Experiments are conducted using an Nvidia GeForce RTX 2080 TI Graphics Card on Mini-ImageNet. The running time of RAR grows linearly with respect to the number of replay iterations. Hence, compared to vanilla experience replay, RAR requires more running time due to the multiple iteration design. However, RAR is still much more computationally efficient compared with offline CL with multiple epochs. An interesting future research direction is to study how to dynamically adjust the iteration number and augmentation strength of RAR to balance the trade-off of accuracy and running time.

### D.5.2 Running time of RL

RL-based hyperparameter optimization is an online method that does not require repeated running over different hyperparameter choices. Therefore, RL-based HPO is much more computationally efficient than offline hyperparameter selection methods. More specifically, the hyperparameter search space in our problem is 100 with 5 augmentation strength levels and 20 memory iteration numbers. Grid search would need to run the OCL algorithm 100 times while RL only needs to run it once.

Table 6: Running time of ER and ER-RAR using an Nvidia GeForce RTX 2080 TI Graphics Card on Mini-ImageNet. The offline setting employs 50 epochs of training.

| | RUNNING TIME (S) | ACCURACY (%) |
|---|---|---|
| ER | $277 \pm 19$ | $20.0 \pm 0.8$ |
| RAR ($K = 5$) | $1383 \pm 4$ | $29.1 \pm 0.9$ |
| RAR ($K = 10$) | $2,345 \pm 4$ | $30.4 \pm 1.3$ |
| R-ER ($K = 10$) | $2,236 \pm 1$ | $17.8 \pm 0.6$ |
| ER-OFFLINE ($E = 50$) | $10,878 \pm 31$ | $20.4 \pm 0.6$ |
| RAR-RL ($K = 19.6$) | $18,037 \pm 71$ | $32.1 \pm 1.0$ |

Nevertheless, the training of RL agents indeed introduces extra computation. In practice, we observe the running time of RL-RAR is about two times slower than that of RAR, as shown in the Table 7.

Table 7: Running time with and without RL-based hyperparameter optimization using an Nvidia GeForce RTX 2080 TI Graphics Card on CIFAR100.

| | RUNNING TIME (S) | ACCURACY (%) |
|---|---|---|
| RAR ($K = 10$) | $1294 \pm 255$ | $27.3 \pm 0.3$ |
| RAR ($K = 20$) | $3499 \pm 406$ | $27.3 \pm 0.5$ |
| RL-RAR ($K = 19.8$) | $6137 \pm 34$ | $29.2 \pm 0.3$ |

### D.6 Offline Continual Learning

Although this paper is mostly focused on online continual learning, some of the analysis and discussion are also of independent interest to offline continual learning. Specifically, from a theoretical perspective, the empirical risk minimization of offline rehearsal is shown in Proposition 2 in Section A and its augmented risk is shown in Proposition 3. Two conclusions can be drawn from this analysis. First, the risk of memory overfitting in offline rehearsal is also related to the problem characteristic, the ratio $\lambda$ between task size and memory size. Second, augmentation can help with offline rehearsal since the orbit-averaging operation in the augmented empirical risk can reduce both the model variance and generalization error. From an empirical perspective, Table 8 shows the performance of offline ER with and without augmentation in four datasets. We use 50 epochs and a memory size of 2000. For all four datasets, offline ER with augmentation achieves a significant performance gain over offline ER without augmentation. More interestingly, compared to datasets with a small $\lambda$ (e.g., CLRS with $\lambda = 1.125$) we observe that the datasets with a higher task-to-memory size ratio (e.g., CORE50 with $\lambda = 6$) tend to benefit more from augmentation, due to the increased risk of memory overfitting.

Table 8: Performance of offline rehearsal with and without augmentation.

|  | SEQ-CIFAR100 | SEQ-MINI-IMAGENET | CORE50-NC | CLRS25-NC |
|---|---|---|---|---|
| OFFLINE ER W/O AUG | $17.0 \pm 0.6$ | $20.4 \pm 0.6$ | $30.2 \pm 1.4$ | $33.5 \pm 1.6$ |
| OFFLINE ER W/ AUG | $28.3 \pm 0.6$ | $32.6 \pm 0.1$ | $44.5 \pm 1.3$ | $35.7 \pm 0.8$ |
| GAINS | $11.3 \uparrow$ | $12.2 \uparrow$ | $14.3 \uparrow$ | $2.2 \uparrow$ |