# OpenReview forum: "A simple but strong baseline for online continual learning: Repeated Augmented Rehearsal"
_NeurIPS.cc/2022/Conference — NeurIPS 2022 Accept_

### Official Review · Reviewer_enAz · 2022-07-05

**Rating:** 5
**Confidence:** 3
**Soundness:** 3 good
**Presentation:** 3 good
**Contribution:** 2 fair

**Summary:**

The paper addresses a rehearsal-based approach for online continual learning. It gives theoretical insights on overfitting and unfitting problem in rehearsal-based approaches. Inspired by the investigation, the paper proposed a simple, novel method called repeated augmented rehearsal.

**Questions:**

When the RL agent is removed, what is recommended to keep a good performance?

**Limitations:**

No experiments on large-scale datasets such as ImageNet-1K.

**Strengths And Weaknesses:**

Strengths:

- strong theoretical analysis of rehearsal-based methods from the perspective of empirical risk minimization.
- the experimental results support the effectiveness of the suggested method.

Weaknesses:
- No experiments on large-scale datasets such as ImageNet-1K.

---

> ### Author Response · Authors · 2022-08-02
> **Author Response to Reviewer enAz**
>
> Thank you for your valuable comments.
>
> **Q1: Large-scale experiment on ImageNet**
>
>
> **R1:** Thank you for your suggestion. We conducted additional continual learning experiments on ImageNet-1k. ImageNet-1k is split into 10 tasks and each task contains 100 classes.
> The training dataset of 10 tasks contains 1,281,167 images in total. Considering the task size, we employ a memory size of M= 20k (with a task-to-memory size ratio of $\lambda=6.4$ ) and a memory size of  M=100k (with a task-to-memory size ratio of $\lambda=1.28$). These choices are similar to Seq-CIFAR100 ($\lambda=1.25$) and CORE50 ($\lambda=6$) with a 2k memory. For RAR method, we use 10 memory iterations with RandAugment with P=1 and Q=14.
>
> We randomly crop the images to size 224x224, and use the ResNet-18 architecture for training on ImageNet-1k with a single epoch. An incoming batch size of 32 and a memory batch size of 32 are used. The results are shown in Rebuttal Table 5.
>
> **Rebuttal Table 5.** Accuracy (%) of ER and RAR for ImageNet-1k.
>
> |  ImageNet-1k    | 20k | 100k|
> |-----------|----------------|----------------|
> | ER        | 2.1 $\pm$ 0.3  |8.7 $\pm$ 0.5 |
> | ER-RAR    | 15.1 $\pm$ 0.4  |34.7 $\pm$ 0.1 |
>
> We observe that repeated augmented rehearsal (RAR) is effective in this large-scale online continual learning problem and improves the vanilla rehearsal from 2.1\% to 15.1\% with a 20k memory and 8.7\% to 34.7\% with a 100k memory.
>
> The discussion in (Buzzega et al., 2020) suggests that to deal with complex continual learning datasets, it is necessary to employ multiple epochs of training with offline CL to avoid the underfitting problem present in online learning. Our results show that RAR can greatly improve online rehearsal for large-scale CL problems. Although several offline CL algorithms have been evaluated on ImageNet-1k, to our knowledge, our result is the first attempt to apply online CL to this problem. We aim to investigate other online rehearsal-based methods on ImageNet-1k in future work.
>
> We have added these results in Appendix D.4 in the revised paper.
>
>
> **Q2: RAR performance without RL**
>
> **R2:** Our paper proposes to use RL to tune the two key hyperparameters of RAR: the number of memory iterations $K$ and the strength of augmentation $P$. We present results for different combinations of values for these hyperparameters in the ablation study in Section 6.4 and find RAR is quite robust to the choice of values. Multiple difference choices lead to a large performance boost over ER (as indicated by the dotted line and the solid line in Figure 4). Specifically, across four datasets, we observe that a relatively large number of memory iterations ( $K=5\sim20$) and a medium level of augmentation ( $P=1$ or $P=2$) generally work well. When using RL to select the hyperparameters, we observe a further performance improvement, but without RL, employing $K=10$ and $P=1$ on all four datasets also leads to a significant performance boost (9\%-19\%) across the four datasets, as shown in Table 1.

---

### Official Review · Reviewer_ga23 · 2022-07-05

**Rating:** 7
**Confidence:** 4
**Soundness:** 4 excellent
**Presentation:** 3 good
**Contribution:** 3 good

**Summary:**

The authors first point out an underfitting vs overfitting dilemma in online continual learning (OCL) with experience replay (ER) on the data distribution and on the samples in the replay buffer, respectively.

They show that the amount of iteration on an incoming batch in Repeated ER helps with the underfitting of the data at the expense of more overfitting on the buffer. This is both proved theoretically and empirically.

The authors propose to combine data augmentation with repeated ER, coined repeated augmented rehearsal (RAR). The motivation for the data augmentation is that it should reduce the buffer's overfitting.

The authors also point out the open problem of Hyperparameter Tuning in OCL. Accordingly, they propose an RL-based hyperparameter tuning algorithm that adapts the hyperparameter in an online manner.

The method is then empirically validated against relevant ER baselines on relevant benchmarks. The paper includes useful ablations and analyses of the results.



**Questions:**

None for now

**Limitations:**

The main limitation I see is that the method requires prior knowledge of data augmentation techniques, which makes it less general than other OCL methods.
Could someone w/o prior knowledge in e.g. audio, RL, use the same RandAugmentatiomn technique? If not, a general protocol to tackle different modalities would be useful.

**Strengths And Weaknesses:**

Theorem 1 is helpful in showcasing how different OCL problem characteristics will impact the underfitting/overfitting dilemma, specifically the data to buffer size ratio.

The repeated rehearsal analysis is useful for the field.
The loss landscape analysis helps the reader to grasp the increasing overfitting problem as the number of iterations (k) increases.
The regularization effect of the incoming data is also something useful, and contrary to the aforementioned observation, I have never given this one previous thoughts.

The method (RAR) is simple and effective.

I also appreciate that the others addressed the open problem of hyperparameter-tuning in OCL with an adaptive RL approach. Its adaptive feature could be particularly useful in OCL with very long task sequences.

The plasticity/stability was particularly interesting.

---

> ### Author Response · Authors · 2022-08-02
> **Author Response to Reviewer ga23**
>
> Thank you for your valuable comments. We have added a discussion of the limitations of data augmentation in Section 7. We agree that applying  RAR in non-visual domains may require other domain-specific augmentation techniques, like those proposed by (Yarats et al., 2020) for RL and (Park et al., 2019) for audio input. We are interested to investigate their effectiveness in future work. In the revised paper, we also add experiments and discussion in Section 6.2 to explore other more general and orthogonal ideas to deal with the overfitting-underfitting dilemma, like reweighting the ER memory loss with a hyperparameter or using logits-based distillation memory loss instead of the cross entropy memory loss. However, the results seem to suggest that even with these advanced rehearsal loss designs, augmentation (in conjunction with repeated replay) still plays an important role in obtaining good performance with online CL.
>
> Yarats, Denis, Ilya Kostrikov, and Rob Fergus. "Image augmentation is all you need: Regularizing deep reinforcement learning from pixels." International Conference on Learning Representations (ICLR). 2020.
>
> Park, Daniel S., et al. "SpecAugment: A Simple Data Augmentation Method for Automatic Speech Recognition." Proc. Interspeech 2019 (2019): 2613-2617.

---

### Official Review · Reviewer_EmdQ · 2022-07-11

**Rating:** 7
**Confidence:** 4
**Soundness:** 3 good
**Presentation:** 4 excellent
**Contribution:** 3 good

**Summary:**

The paper studies the effect of data augmentation in online continual learning. It proposes *repeated augmented rehearsal*, a simple method that adds data augmentation when using multiple iterations on the same batch in online continual learning.

**Questions:**

1. Did you consider the (partial) remedy of reweighting the memory loss term in Eq. (2)? If yes, why did you not include this option in your method? If no, do you see any reasons not to use this simple (partial) remedy in conjunction with data augmentation?

2. Why did you choose to focus exclusively on the online CL setting? Do you think it is of great practical relevance? Would you agree that some of the main questions the paper is raising are also relevant in the offline CL setting?

**Limitations:**

I think the limitations of the paper could be discussed more honestly. Some of the consideration around the "regularization effect" of incoming data are mostly anecdotal (e.g., Figures 1 and 2) but that is not clearly stated. Also, data augmentation seems to be limited to image data, which could have been mentioned in the paper.

**Strengths And Weaknesses:**

### Strengths

1. I consider the simplicity of the proposed approach a major strength. Continual learning research suffers from a flood of overly complex methods without proper ablation studies. Revisiting the seemingly simple baselines, with proper ablation studies, is an important contribution!

2. The empirical evaluation is of high quality, includes most relevant baselines and competing methods, and provides a number of ablation studies. (A slight caveat is that the same architecture, ResNet18, is used on all datasets.)

3. The results of the proposed method in the online continual learning setting are very convincing. RAR provides significant performance boosts across the board and can be combined with any rehearsal-based CL method.



### Weaknesses

1. I think this paper has one major flaw. It assumes the ER formulation in Eq. (2), which weights loss contributions from memory and incoming batch equally. One can easily upweight or downweight the memory by reweighting the first gradient term in Eq. (2), which controls the memory overfitting issue and (if done properly) can also remove the undesirable dependence on the task size. Therefore, in some sense the paper highlights a problem of vanilla ER, but neglects the most straight-forward (partial) remedy. Since the goal of the paper is to revisit a simple baseline, this constitutes a major shortcoming in my opinion.

2. I find it somewhat regrettable that the authors chose to exclusively focus on the *online* continual learning setting, which seems of limited practical relevance to me. (One can always buffer data from a stream into mega-batches and run an "offline" continual learning algorithm on the resulting "pseudo-tasks".) The key questions the paper studies, the effect of data augmentation in rehearsal-based continual learning, is just as relevant in the "offline" CL setting.

3. Finally, I want to register that the resulting method is of limited novelty, simply combining repeated ER with data augmentation. (Personally, I don't think that is a big caveat, but should be mentioned.) I would also add that augmentation can only ever be a partial remedy. It can ameliorate the memory overfitting issue and is certainly useful in practice. But, in some sense, it will never tackle the fundamental problem.

4. Minor Comments

    - The notation in Eq. (2) is sloppy, since the examples are indexed with $i$, but this is not the summation index of the sum. Why not just $\sum_{x, y \in \mathcal{B}_t}$? This would also avoid the currently overloaded notation for data points from memory and incoming batch.
    - Calling Theorem 1 a Theorem is a bit much in my opinion. I would go with a more modest "Proposition", but I guess that is a question of personal taste.

### Rating

I explicitly commend the effort to revisit simple, principled baselines. The paper is well-written and the empirical evaluation is sound. Unfortunately, I have to recommend rejection for this paper. I think omitting the option to reweight the memory loss (Weakness 1) is a fundamental shortcoming when trying to study the issue of memory overfitting in continual learning.

### Update after rebuttal

The authors' rebuttal has resolved my main concern regarding the reweighted ER baseline. I have increased my score accordingly.

---

> ### Author Response · Authors · 2022-08-02
> **Author Response to Reviewer EmdQ**
>
> Thank you for your constructive comments. We address your concerns point by point:
>
> **Q1: Reweighting hyperparameter for memory loss**
>
> **R1**: Thanks for suggesting the idea of reweighting the memory loss. Corresponding experiments that we have performed now indicate that this approach is orthogonal to the algorithmic modifications proposed in RAR and can be combined profitably with them. We plot the performance of ER and RAR for different values of the reweighting hyperparameter ($\alpha \in [0.1,0.3,0.5,0.7,1.0,2.0]$) in [Fig 10](https://anonymous.4open.science/r/rebuttal_results-B863/Figure_10_v2.PNG), Appendix D3 in the revised paper.
> A key observation is that similar to vanilla ER, reweighted ER (ER-rw) also significantly benefits from RAR, as ER-rw-rar greatly improves over ER-rw (see the red line and blue line in Fig. 10) for all four datasets.
>
> The performance using the optimal reweighting values is reported in Rebuttal Table 4. ER-rw-RAR leads to significant performance gain over ER-rw (+8.5\% CIFAR100, +10.5\% Mini-ImageNet, +24.9\% CORE50-NC,+10.2\% CLRS25). This result suggests that the RAR technique can be applied to reweighted ER loss to improve the performance of online rehearsal.
>
> Another observation in Fig. 10 is that ER-rw-rar can lead to performance improvement over ER-RAR ($\alpha=1.0$) for some datasets (CIFAR100 and CORE50) but not others (Mini-ImageNet and CLRS).
> This implies that reweighting the ER loss function can help with memory overfitting in some cases. However, the interplay of repeated rehearsal and augmented rehearsal seems to play a more important role in defying the dilemma of overfitting locally but underfitting globally in online CL.
>
> In the revised paper, we have added a paragraph in Section 6.2 to discuss these new results and the effect of RAR on ER-rw.
>
> **Rebuttal Table 4.** Results of reweighting the ER loss.
>
> |           | CIFAR100   | MINI-IMAGENET | CORE50-NC   | CLRS25-NC |
> |-----------|----------------|----------------|----------------|----------------|
> | ER-rw     | 22.0 $\pm$ 1.3 | 20.0 $\pm$ 0.8  | 24.9 $\pm$ 1.8 | 18.4 $\pm$ 1.5 |
> | ER-rw-RAR | 30.5 $\pm$ 0.3 | 30.5 $\pm$ 0.3 | 47.8 $\pm$ 1.9 | 28.6 $\pm$ 2.7 |
>
>
>
> **Q2: Focus on online CL**
>
> **R2:** We indeed believe that online CL is of great practical relevance. First, we agree with De Lange et al (2021) and Buzzega et al (2020) that continual learning methods are expected to be employed in a general setting where "*task boundaries can be blurred and the domain and class distributions can shift either gradually or suddenly*"(Buzzega 2020). This defines a General Continual Learning (GCL) setting without the notion of task, class, or domain. Offline continual learning relies on task boundaries to populate the buffer to perform offline training and is a relaxation of this GCL setting (De Lange 2021). We believe the study of online CL is the key to moving towards general continual learning. Second, online CL can support timely updates in a practical interactive system, as it can perform immediate updates as soon as it receives a new batch of data, while offline CL needs to wait until the collection of a whole task's datasets. Third, online CL has a lower storage cost than offline CL, as it does not need to store the whole dataset of a task. Thus, online CL can be used for applications with a very limited storage budget.
>
> - De Lange et al. "A continual learning survey: Defying forgetting in classification tasks." IEEE Trans. PAMI 44.7 (2021): 3366-3385.
> - Buzzega, Pietro, et al. "Dark experience for general continual learning: a strong, simple baseline." NeurIPS (2020): 15920-15930.
>
> We agree that the effect of augmentation is also relevant to offline CL. In Appendix A, we analyze the loss function of *offline* CL (proposition 2) and the augmented loss function for offline CL (proposition 3). Since the focus of our paper is online CL, we consider presenting results for offline CL out of scope.
>
> Compared to offline rehearsal, online rehearsal has faced particular challenges in tackling complex CL datasets due to the single-epoch constraint. While offline rehearsal can solve the underfitting issue by increasing the number of epochs (to 50-200 epochs), online rehearsal cannot simply raise the number of memory iterations, as discussed in Section 3: repeated rehearsal may not necessarily lead to better performance, due to the biased SGD.
> Our work tries to analyze the internal workings of online rehearsal and identifies the fundamental challenge that it faces as the dilemma between overfitting locally and underfitting globally. We believe solving this dilemma with RAR can help with the application of online rehearsal methods in complex datasets and moving towards general continual learning. Therefore, this paper is focused on online CL.
>
> **Other changes**
> - added a line in Section 7 to clarify the limitations of data augmentation.
> - modified Eq.2 and other equations.
> - changed "Theorem 1" to "Proposition 1".

---

> > ### Comment · Reviewer_EmdQ · 2022-08-05
> > **Great revisions!**
> >
> > Thank you for engaging so thoroughly with my review and for these extensive additional experiments and revisions. I'm sure this was a substantial amount of work in such a short time and I commend you for it!
> >
> > ### Re Q1
> > Thank you for running these experiments. They potentially address my main concern, since the results seem to show that balancing the two loss contributions can help a little bit but still benefits greatly from data augmentation.
> >
> > I have a follow-up question on one detail: How exactly did you implement reweighted ER? Is it
> > $$ \alpha \frac{1}{B_t} \sum_{x, y\in B_t} \nabla L(f_\theta(x), y) + \frac{1}{B^M_t} \sum_{x, y\in B^M_t} \nabla L(f_\theta(x), y), \quad \alpha > 0$$
> > or
> > $$ \alpha \frac{1}{B_t} \sum_{x, y\in B_t} \nabla L(f_\theta(x), y) + (1-\alpha) \frac{1}{B^M_t} \sum_{x, y\in B^M_t} \nabla L(f_\theta(x), y), \quad \alpha\in [0, 1]$$
> > I would strongly favor the second option, because the first option changes the overall scale and would necessitate adjusting the step size. Using the first option with the same step size for every value of $\alpha$ could lead to wrong conclusions.
> >
> > The exact description of reweighted ER should also go in the final version of the paper.
> >
> > ### Re Q2
> >
> > Let me start by saying that you chose to focus on online CL for this paper and I don't expect you to change this. But I still disagree with your assessment of online vs offline CL and I think the restriction to the online setting unnecessarily limits the scope of your contribution.
> >
> > The difference between the offline and online CL has nothing to do with the notion of tasks or domains. Offline CL can be completely agnostic to the notion of task, domain or class. A new dataset $D_t$ arrives; it might correspond to a distinct "task" or "domain", but it need not be so. In offline CL, you are allowed to train on this dataset for multiple epochs before moving on to the next one. Online CL makes the (imo artificial) restriction that you are only allowed to do a single pass over this dataset. If you have $D_t$ available in its entirety, why would this be a sensible restriction? If $|D_t|$ is so huge that you run into memory issues, you can chunk it up into smaller (but still much larger than minibatches) subsets and run an offline CL algorithm on those. Of course, if your use case provides datasets $D_t$ of sizes that are roughly the size of minibatches then you effectively end up in an online setting.

---

> > > ### Author Response · Authors · 2022-08-07
> > > **Author response to follow-up questions and comments of Reviewer EmdQ**
> > >
> > > Thank you for your positive feedback and valuable suggestions!
> > >
> > > **Implementation of Reweighted ER**
> > >
> > > The gradient of reweighted ER is implemented similar to Eq 1, where $\alpha$ denotes the ratio between the weight of memory loss and incoming loss. We agree that the performance ER-rw-Eq1 is influenced by both the reweighted hyperparameter $\alpha$ and the step size. We run new experiments using Eq 2, where the ratio between memory loss weight and incoming loss weight is $\alpha/(1-\alpha)$. The new results are shown in Rebuttal Table 4.1 and [Rebuttal Fig 1](https://anonymous.4open.science/r/rebuttal_results-B863/Fig10_v3.PNG) with $\alpha \in [0.1,0.3,0.5,0.7,0.9]$ and $\alpha=0.5$ denoting equal weighting of memory and incoming losses. Similar to previous findings, we observe that 1) reweighted ER with RAR approach, i.e. ER-rw-RAR (Eq2), leads to significant performance gains over reweighted ER (Eq2) for four datasets. This result confirms the effectiveness of RAR with reweighted ER loss; 2) Reweighted ER-RAR (Eq2) improves vanilla ER-RAR in CIFAR100 and CORE50 but not for Mini-ImageNet and CLRS25.
> > >
> > > We will clarify the implementation of the reweighted ER and add the new results in the revised paper.
> > >
> > > Implementation of ER-rw (**Eq1**)
> > >
> > > $g_{t}= \frac{1}{|B_t|} \sum_{x,y \in  B_t} \nabla{L}(f_\theta({x}),y) +\alpha\frac{1}{|B^M_t|} \sum_{x,y \in B^M_t} \nabla {L}(f_\theta({x}),y), \alpha>0$
> > >
> > > Implementation of ER-rw ( **Eq 2**)
> > >
> > > $g_{t}=(1-\alpha) \frac{1}{|B_t|} \sum_{x,y \in  B_t} \nabla{L}(f_\theta({x}),y) +\alpha\frac{1}{|B^M_t|} \sum_{x,y \in B^M_t} \nabla {L}(f_\theta({x}),y), \alpha \in (0,1)$
> > >
> > >
> > > **Rebuttal Table 4.1** Further investigation of Reweighted ER with Eq 2 implementation.
> > >
> > > |           | CIFAR100   | MINI-IMAGENET | CORE50-NC   | CLRS25-NC |
> > > |-----------|----------------|----------------|----------------|----------------|
> > > | ER        | 19.0 $\pm$ 0.6 | 20.0 $\pm$ 0.8 | 24.0 $\pm$ 2.0 | 18.7 $\pm$ 1.6 |
> > > | ER-RAR    | 27.8 $\pm$ 0.5 | 30.0 $\pm$ 0.9 | 39.3 $\pm$ 1.4 | 28.6 $\pm$ 2.7 |
> > > | ER-rw-Eq2     | 21.0 $\pm$ 1.0 | 20.1 $\pm$ 0.8  | 24.6 $\pm$ 0.6 | 19.2 $\pm$0.6 |
> > > | ER-rw-RAR-Eq2 | 30.8 $\pm$ 0.1 | 30.4 $\pm$ 1.3 | 45.3 $\pm$ 2.2 | 28.6 $\pm$ 2.7 |
> > >
> > > **Online and Offline Continual Learning**
> > >
> > > We agree that in the cases where a dataset $D_t$ is already available in advance, offline CL algorithms can be employed to train the dataset $D_t$ for multiple epochs. However, deep neural networks usually require a large dataset $D_t$ and extensive training to work well.  We believe online CL is particularly useful in dealing with continuous non-stationary data streams generated by online applications, e.g. social media applications or interactive robotic systems. Since it does not need to wait for the collection of a (large) dataset $D_t$ ( as mentioned by the reviewer, online CL can be regarded as using a tiny dataset $D_t$ of batch size, e.g. 10 samples in our experiments), it can make a timely personalized update to the system as long as it receives some user interation data, potentially facilitating a better user experience.
> > >
> > > In addition, considering the applications with limited memory, an interesting research question is the memory allocation issue: which leads to better CL performance, to allocate more memory space to store rehearsal samples, or spend some space to store incoming data $D_t$ (as in the offline CL)?
> > >
> > > We plan to discuss these questions and the effect of augmentation for offline CL in the extended version of the paper.

---

> > > > ### Comment · Reviewer_EmdQ · 2022-08-09
> > > > **Good paper!**
> > > >
> > > > **Reweighted ER:** Thanks for clarifying the detail regarding reweighted ER. This resolves my main concern and I will increase my score accordingly.
> > > >
> > > > **Online vs offline CL:** I guess we have to agree to disagree regarding the relative importance of the online vs offline CL setting. I'm not saying that there are _no_ use cases for online CL (you mentioned some in your rebuttal) but that the offline CL setting is much more relevant in practice. For the final version of the paper, I think it would be great if you could try to comment on if/how your findings are relevant to the offline CL setting as well.

---

> > > > > ### Author Response · Authors · 2022-08-09
> > > > > **Thanks for the positive feedback**
> > > > >
> > > > > Thank you very much for providing the insightful discussion and helpful comments. We highly appreciate that! We will include a discussion on how the findings in this work may also be relevant to offline CL in the final version of the paper.

---

### Official Review · Reviewer_P1CH · 2022-07-19

**Rating:** 6
**Confidence:** 4
**Soundness:** 3 good
**Presentation:** 2 fair
**Contribution:** 2 fair

**Summary:**

This paper presents an online Experience replay (ER) based continual learning technique to mitigate forgetting of the past data. The paper considers a repeated rehearsal setting that allows training for multiple steps on each batch of data (previously proposed by Aljuni et al. 2019). In this repeated rehearsal setting, the paper studies the source of replay-memory overfitting.

The paper proposes a formulation of ER loss function that captures not only the dependence on the number of samples from the current and past tasks (shown in previous work), but also accounts for the ratio of incoming data size and the replay-memory size (novel contribution claim). When the relative size of replay-memory is smaller as compared to the size of the incoming task, memory overfitting can occur. As the number of previously-seen tasks increases, the memory overfitting worsens.

Further, the paper analyzes the training process for any given task and observes that repeated rehearsal leads to a quick drop in the training loss of the incoming data. Therefore, during the later iterations of training, the memory loss dominates the incoming data loss, and reduces the regularization effect of incoming data. The authors describe this phenomenon as an underfitting-overfitting dilemma – where repeated rehearsal can reduce underfitting of incoming data but at the same time can cause overfitting of the replay-memory data.

In order to solve the over-fitting problem, the authors propose data-augmentation of both the incoming data and the replay-data during mini-batch training. A RL-based HPO scheme is employed during training to tune the two key hyper-parameters – number of repeated rehearsals and strength of augmentation.

The proposed method is evaluated on a suite of computer-vision tasks and is shown to outperform ER baselines.


**Questions:**

1. The training loss for the incoming batch in Figure 2 (a) goes down very quickly. Why do you not observe overfitting on the incoming training batch during repeated rehearsal (say k=10)?
2. It is not clear to me what is the reward function used for RL-based HPO.
3. What is the compute overhead of RL-based HPO?


**Limitations:**

The authors mention some limitations of RL-based HPO but do not specify the the compute overhead.

**Strengths And Weaknesses:**

Strengths:
1. The paper analyzes the replay-memory overfitting problem in detail.
2. The paper combines existing techniques of repeated rehearsal with data-augmentation to alleviate overfitting on replay-memory data.
3. The paper performs ablation studies to understand the effect of repeated rehearsal and augmentation.
4. Use of RL-based HPO during online learning is an interesting direction.
5. The paper is mostly well-written.

Weaknesses:
1. The paper does not cite and compare against Dark ER [Buzzega et al. 2020]. This prior work also employs data-augmentation and is also applicable to the online learning setting.
2. Previous ER formulations often use an additional hyper-parameter that balances the loss between incoming data and the new data. Not sure why this hyper-parameter is missing in the problem formulation.
3. I could not find any example of the type of augmentations used in the main paper.
4. Writing improvements:
   (a) It is difficult to follow Figure 1 due to smaller font and lack of enough details in the figure caption.
   (b) Typo in Line 213.

---

> ### Author Response · Authors · 2022-08-02
> **Author Response to Reviewer P1CH**
>
> Thank you for your constructive comments. We address your concerns point by point as below:
>
> **Q1: Dark ER**
>
> **R1**: Thanks for suggesting the Dark ER method.  DER employs a distillation memory loss instead of cross-entropy loss. Buzzega et al. reported difficulties in applying DER with single-epoch training on complex datasets such as CIFAR-10 and ended up using offline CL with 50 epochs.
>
> Our new results of DER for online CL  also suggest that DER (with or without augmentation) does not work well in an online setting, as it leads to worse performance than ER. Interestingly, when combined with RAR, the performance of DER is greatly improved for all four datasets. This suggests that even with advanced rehearsal loss design, RAR is important to achieve good performance for online rehearsal.
>
> We have added these results in Table 1 in the revised paper and also added a paragraph in Section 6.2 to discuss this finding.
>
> **Rebuttal Table 1.**  Accuracy of DER and ER-rw.
>
> |           | CIFAR100   | MINI-IMAGENET| CORE50   | CLR      |
> |-|-|-|-|-|
> | DER w/aug   | 8.4 $\pm$ 0.6  | 11.8 $\pm$ 0.5 |23.8 $\pm$ 0.6  | 11.8 $\pm$ 2.6 |
> | DER w/o aug  | 9.1 $\pm$ 1.4  | 12.0 $\pm$ 1.2 |24.6 $\pm$ 1.2  | 14.0 $\pm$ 0.5 |
> | DER-RAR   | 30.0 $\pm$ 1.2 | 26.2 $\pm$ 0.4 |37.7 $\pm$  1.4 | 28.4 $\pm$ 3.2 |
> |  | |  | |  |
> | ER-rw     | 22.0 $\pm$ 1.3 | 20.0 $\pm$ 0.8  | 24.9 $\pm$ 1.8 | 18.4 $\pm$ 1.5 |
> | ER-rw-RAR | 30.5 $\pm$ 0.3 | 30.5 $\pm$ 0.3 | 47.8 $\pm$ 1.9 | 28.6 $\pm$ 2.7 |
>
>
> **Q2: Weighting the components of the loss**
>
> **R2**: New results using different reweighting hyperparameter  ($\alpha \in [0.1,0.3,0.5,0.7,1.0,2.0]$) for memory loss are shown in [Fig 10](https://anonymous.4open.science/r/rebuttal_results-B863/Figure_10_v2.PNG) of Appendix D3.
>  Similar to vanilla ER, reweighted ER (ER-rw) also benefits from RAR, as ER-rw-rar greatly improves on ER-rw (as indicated by the red line and blue line in Fig. 10). This suggests RAR can be combined with the reweighted ER loss function to improve online CL performance.
>
> **Q3: Rapid decrease of loss on incoming data in Fig 2 (a) and overfitting risk**
>
> **R3:** Relative to data in the memory, there is a lower risk of overfitting incoming data because incoming samples are only seen once by the CL model while memory data can be repeatedly sampled and accessed. For repeated ER, when the number of iterations K is increased from 1 to 5,  the performance on the current task (as measured by plasticity) is significantly increased (see the solid line in Fig 3). This shows the training of the current task benefits from more iterations and implies the incoming data is not overfitted. In contrast, the performance on past tasks (as measured by stability) is greatly decreased with an increased number of iterations, suggesting the overfitting of memory data.
>
> We also find overfitting of incoming data is possible for large values of K. With K = 5 to 20, the current task's performance (plasticity) starts to decrease. Interestingly, with RAR, plasticity continues to improve even with large K values, indicating that augmentation decreases the risk of overfitting incoming data.
>
>
> **Q4: Reward of RL-based HPO**
>
> **R4**: Due to the page limit, the details of the reward design are presented in Appendix C.2. To achieve online HPO without external validation data, we define reward as  $r=|A_M-A^*|$, where  $A_M$ is the current memory accuracy and $A^*$ is a chosen target memory accuracy. We added a line in Section 5 to clarify this.
>
> **Q5: Overhead of RL-based HPO**
>
> **R5:** RL-based HPO is an online method that does not require repeated running to evaluate different hyperparameter choices. Thus, RL-based HPO is more computationally efficient than offline hyperparameter selection methods. More specifically, the hyperparameter search space in our problem is 100, with 5 augmentation strengths and 20 settings for the number of memory iterations. This means grid search would need to run the OCL algorithm 100 times while online HPO using RL only needs to run it once.
>
> Nevertheless, training RL agents indeed introduces extra computation. In practice, we observe the running time of RL-RAR is about two times slower than that of RAR, as shown in the table below. We have added Section D.8.2 in the appendix to discuss the computational efficiency of RL-based HPO.
>
> **Rebuttal Table 3.** Running time on CIFAR100.
>
> |           | Running Time (s)   | Accuracy (%)|
> |-|-|-|
> | RAR (N=20)     |3499  $\pm$ 406| 27.3  $\pm$ 0.5|
> | RL-RAR (N=19.8)| 6137 $\pm$ 34 | 29.2 $\pm$ 0.3 |
>
>
> **Q6: Augmentation type**
>
> **R6:** We use an auto augmentation method, RandAugment (Cubuk 2020). It randomly selects P number of augmentation operators from a set of 14 operators for each image. We clarified this in Appendix C.1. We also report results using a fixed augmentation type (crop, flip, color jitter, gray) in Appendix D.6
>
> **Other changes**
>  * Enlarged texts in Fig 1; added more caption description.
>  * Corrected the typos

---

### Author Response · Authors · 2022-08-02
**Summary of changes in the revised paper**

We thank all reviewers for their valuable comments. We have revised the paper in response to these comments and have submitted a revised version. The main changes are:

- Added results for reweighted memory loss (ER-rw) in Table 1.
- Added results for the DER method in Table 1.
- Added a paragraph in Section 6.2 to discuss the effectiveness of RAR with modified rehearsal loss.
- Added results of a large-scale experiment with ImageNet in Table 5 of Appendix D4.
- Added the running time of RL-based HPO in Table 8 of Appendix D8.2.
- Modified Section 7 to further clarify the limitations of augmentation.
- Added a line in Section 5 to clarify the reward design of RL.

Minor changes include:

* Added more description in the caption of Figure 1 and increased the font size in the figure.
* Modified the symbols used in the equations to avoid overloaded notations.
* Corrected the typos.

---

### Author Response · Authors · 2022-08-09
**Looking forward to your reply**

Dear Reviewers,

Thanks a lot for your efforts in reviewing this paper. We tried our best to address all the mentioned concerns/questions and revised our paper in response to the comments.

In the latest revision of the paper, we include minor changes (shown in red) to clarify the reweighted ER implementation in Table 1 and Figure 10 in Appendix D.3. Other revised parts mentioned previously are denoted in blue.

If you have any further questions, please let us know.

Kind regards,
Authors

---

### Author Response · Authors · 2022-08-09
**Thank you for the feedbacks**

We thank the reviewers for the constructive feedback and insightful discussions to help refine the paper.

The main revisions are briefly summarized as follows:
* add a large-scale experiment using *ImageNet-1k*;
* add discussion and comparison experiment with *DER*;
* add discussion and experiments for *reweighted ER*;

Since the rebuttal discussion is about to end soon, if there are any further clarifications required or any other concerns, please let us know. Thank you!

---

### Meta-Review · Area_Chair_uCM9 · 2022-08-28

**Recommendation:** Accept
**Confidence:** Certain

**Metareview:**

This work considers rehearsal-based methods in continual learning and revisits revisits the rehearsal dynamics. Most reviewers praised the analysis of overfitting/underfitting in replay-based methods and the presentation. The proposed repeated rehearsal with data-augmentation was shown to be effective in empirical evaluations and the ablation studies. Finally, the authors addressed on of the main concerns raised by the reviewers, namely the relation to the reweighted ER baseline, during the rebuttal.




**Award:**

No

---

### Decision · Program_Chairs · 2022-09-14

Accept